# Ongoing movement controls sensory integration in the dorsolateral striatum

Roberto de la Torre-Martinez ●[1] ✉, Maya Ketzef ●[1,2] & Gilad Silberberg ●[1,2] ✉

The dorsolateral striatum (DLS) receives excitatory inputs from both sensory and motor cortical regions. In the neocortex, sensory responses are affected by motor activity, however, it is not known whether such sensorimotor interactions occur in the striatum and how they are shaped by dopamine. To determine the impact of motor activity on striatal sensory processing, we performed in vivo whole-cell recordings in the DLS of awake mice during the presentation of tactile stimuli. Striatal medium spiny neurons (MSNs) were activated by both whisker stimulation and spontaneous whisking, however, their responses to whisker deflection during ongoing whisking were attenuated. Dopamine depletion reduced the representation of whisking in direct-pathway MSNs, but not in those of the indirect-pathway. Furthermore, dopamine depletion impaired the discrimination between ipsilateral and contralateral sensory stimulation in both direct and indirect pathway MSNs. Our results show that whisking affects sensory responses in DLS and that striatal representation of both processes is dopamine- and cell type-dependent.

Sensory processing in various cortical regions has been shown to be modulated by motor activity[1–3]. While locomotion reduces the magnitude of auditory responses in auditory cortex[4–6], it enhances visual responses in visual cortex[7–9]. In the somatosensory cortex, tactile responses to whisker deflection are attenuated during active whisking[1,10,11]. These studies show that sensory and motor processes are tightly linked at the cortical level, however, it is not known how such sensorimotor interactions are represented in downstream subcortical regions. The whisker system is an essential sensory apparatus in rodents, utilizing an active sensing framework, in which whisking and tactile sensation interact to represent the location and identity of objects[12]. The dorsolateral striatum (DLS) receives dense monosynaptic excitatory input from primary somatosensory cortex and is involved in sensory representation[13–18]. Striatal medium spiny neurons of both the direct- and indirect pathways (dMSNs and iMSNs, respectively) respond to whisker stimulation[16,19–23]. Importantly, MSNs have distinct representations for contralateral and ipsilateral tactile inputs[16,23], a feature that is lost following dopamine (DA) depletion, as shown in anesthetized mice[23]. The DLS also receives axonal projections from motor cortical regions[14,19,22,24,25]. Such convergence of inputs from both somatosensory and motor

cortices suggests that sensory responses are modulated by motor activity also at the striatal level.

Parkinson's disease (PD) is a neurodegenerative motor disorder associated with the progressive death of midbrain dopaminergic neurons of the substantia nigra *pars compacta*, a region that, among other structures, densely innervates the striatum[26,27]. As a result, PD patients typically present motor deficits such as muscle rigidity, tremor, or bradykinesia[28–30], some of which are caused by abnormal activation of dMSNs and iMSNs[23,31–41]. In addition to these motor impairments, PD patients often exhibit somatosensory symptoms such as deficits in the perception of thermal, pain, proprioceptive, and tactile stimuli, including impairments in bilateral tactile discrimination[30,42–45].

Here we examined whether and how tactile sensory integration is modulated by motor activity in the whisker system, and how such sensorimotor interactions are altered in a PD mouse model. To address these questions, in vivo whole-cell recordings in awake mice were performed in the DLS while bilateral whisker stimulation was delivered in control (DA-intact) and DA-depleted mice. Responses to sensory stimulation were recorded during different behavioral states, from optogenetically identified MSNs. Our results show that the majority of

[1]Department of Neuroscience, Karolinska Institutet, Stockholm, Sweden. [2]These authors jointly supervised this work: Maya Ketzef, Gilad Silberberg.
✉e-mail: roberto.de.la.torre.martinez@ki.se; gilad.silberberg@ki.se

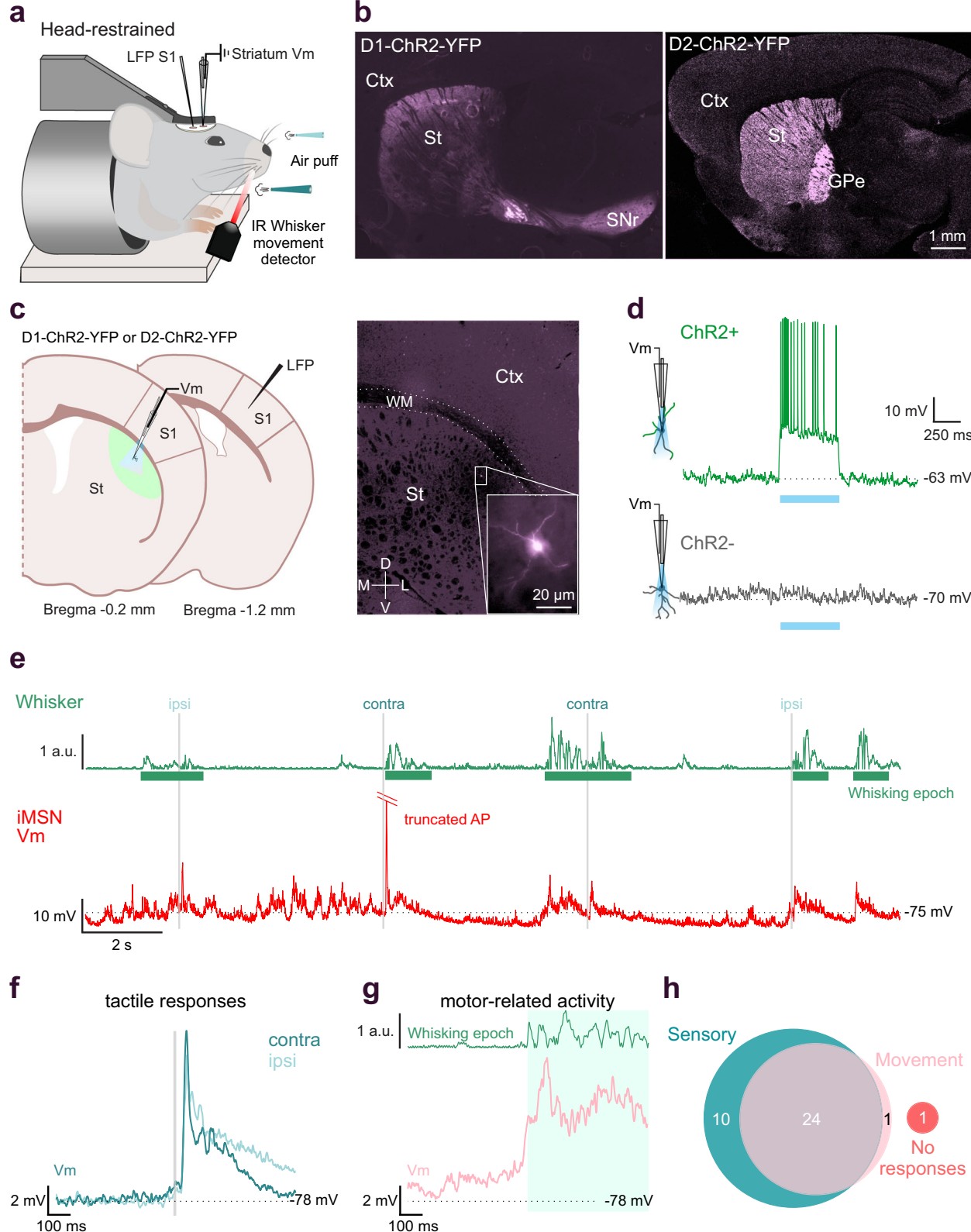

MSNs in the DLS responded to both whisker stimulation and to self-generated whisking. Whisking onset was accompanied with depolarization of MSNs, however, sensory responses to whisker deflection were attenuated during whisking. DA depletion affected both motor and sensory responses in a cell type-specific manner. Our results provide insights into the network mechanisms underlying sensorimotor processing in the basal ganglia and its impairment in PD.

## Results

### In vivo whole-cell recordings from identified MSNs in awake mice

To study how sensory processing in MSNs is affected by motor activity, we performed in vivo whole-cell recordings in the DLS of awake, head-restrained mice, during epochs of quiescence and spontaneous whisking (Fig. 1). Identification of neurons during

**Fig. 1 | Whole-cell recordings from opto-tagged MSNs in the DLS of awake behaving mice. a** Schematic of the experimental setup. Whole-cell recordings were performed in the mouse DLS with the optopatcher for online optogenetic classification, simultaneously with LFP recordings in S1. Whiskers were stimulated by air-puffs delivered independently to ipsilateral and contralateral whiskers, and whisking behavior was monitored using a non-contact infrared LED-photodiode. **b** Opto-tagging of MSNs was obtained using either D1-ChR2-YFP (labeling dMSNs, left) or D2-ChR2-YFP (labeling iMSNs, right) mice (*n* = 15 mice for D1, *n* = 19 mice for D2). Typical projections from dMSNs in D1-ChR2-YFP mice to SNr and from iMSNs to the GPe in the D2-ChR2-YFP mice are evident. Scale bar, 1 mm. **c** Schematic representation showing whole-cell and LFP recording locations from Bregma (left). The image and magnified inset (right) show an example (from 34 independent neurons with similar results) of a biocytin-filled MSN from DLS following an in vivo whole-cell recording in a behaving mouse. The inset shows the same cell in higher magnification. **d** Opto-tagging of MSNs using the optopatcher. Depolarizing responses to photostimulation of a positive cell (ChR2 + ) in D1-ChR2-YFP mouse (top) for the duration of the stimulation. Negative cells (ChR2 − , bottom) did not respond with depolarization to photostimulation. **e** Example of spontaneous membrane potential activity in an opto-tagged iMSN, whole-cell recorded in the DLS of an awake behaving mouse. Membrane potential (Vm, red) of the neuron was recorded simultaneously with measurement of whisker activity (Whisker, green), air puff stimulation indicated in gray (ipsi/contra). **f** Synaptic responses of the neuron showed in **e**, to contra- and ipsilateral whisker deflection. **g** Depolarization of the membrane potential preceded whisker movement in the same neuron showed in **e**. **h** Venn diagram showing the number of MSNs responding to whisker stimulation (turquoise circle) and to spontaneous whisking movement (pink circle) or absence of both (red circle). LFP Local Field Potential, Barrel field somatosensory cortex S1, IR infrared light, SNr Substantia Nigra *pars reticulata*, GPe Globus Pallidus externa, Ctx Cortex, WM White Matter, St Striatum, ChR2 Channelrhodopsin, a.u. arbitrary units.

in vivo patch-clamp striatal recordings is challenging since the electrophysiological properties of dMSNs and iMSNs are not distinct enough to allow for unambiguous classification. To overcome this limitation, we used the "optopatcher"[46] in transgenic mice expressing Channelrhodopsin (ChR2) in either dMSNs or iMSNs (Fig. 1b). We verified the selective expression of YFP-ChR2 in dMSNs or iMSNs by fluorescent labeling of striatonigral and striatopallidal projections, respectively (Fig. 1b, c). This approach allowed us to identify dMSNs and iMSNs in real-time during recordings using focal photostimulation through the patch pipette (Fig. 1d). MSNs expressing ChR2 responded to blue light pulses with immediate and robust depolarization, usually leading to action potential (AP) discharges, whereas ChR2-negative MSNs did not respond to the photostimulation (Fig. 1d). The recorded neurons were filled with biocytin to allow for post-hoc visualization of their location in the DLS and morphological characterization (Fig. 1c). Whisker movements were simultaneously tracked using an infrared LED sensor, allowing the definition of quiescent (Q) and whisking (W) epochs (Fig. 1e). Most of the MSNs recorded in the DLS responded to tactile stimulation (94.44%, *n* = 34/36, Fig. 1f, h), and depolarized prior to whisker-related motor activity (66.66%, *n* = 24/36 Fig. 1g, h). Only two of the recorded neurons did not respond to whisker stimulation, one of which depolarized during spontaneous whisking (Fig. 1h).

## DA depletion reduces activity in dMSNs

To investigate the effects of DA depletion on the membrane properties of striatal MSNs, we obtained whole-cell recordings from DA-intact (control) and DA-depleted awake mice (Fig. 2a, c). DA-depleted mice received a unilateral injection of 6-hydroxydopamine (6-OHDA) in the medial forebrain bundle that produced a nearly complete depletion of DA innervation in the striatum of the injected side (Fig. 2a, b). We verified the absence of DA by immunofluorescence staining for tyrosine hydroxylase (TH) (Fig. 2b). Analysis of whisking activity revealed that mice engaged in spontaneous whisking roughly 15% of the time throughout the recording session, with no differences in the overall whisking duration between control and DA-depleted mice. There were, however, differences in the frequency and duration of whisking bouts, with shorter and more frequent bouts in 6-OHDA lesioned mice (Supplementary Fig. 1). During quiescent periods, no differences were observed in the resting membrane potential of dMSNs (control: −67.18 ± 1.30 mV; 6-OHDA: −67.89 ± 1.15 mV, *P* > 0.05, Fig. 2d) despite a higher input resistance in lesioned mice (control: 120.10 ± 13.59 MΩ; 6-OHDA: 189.90 ± 24.47 MΩ, *P* < 0.05, Fig. 2e). Interestingly, DA depletion significantly reduced the spontaneous firing rate of dMSNs (control: 0.13 ± 0.07 Hz from 5 out of 15 neurons; 6-OHDA: 0 out of 14, *P* < 0.05, Fig. 2f). In contrast, none of the previously mentioned properties changed in iMSNs (*P* > 0.15, Fig. 2d–f). Overall, our results show predominant changes in the membrane properties of dMSNs following DA depletion.

## MSNs depolarize before the onset of motor activity

We next measured the membrane potential dynamics during whisking in a subset of 48 MSNs, 23 in control mice (9 dMSNs and 14 iMSNs), and 26 in DA-depleted mice (9 dMSNs and 17 iMSNs) (Fig. 3). In control mice, dMSNs, and iMSNs showed progressive depolarization that started before whisker movement onset (dMSNs: 99.33 ± 28.84 ms, iMSNs: 90.56 ± 36.60 ms), and peaked after whisking onset (dMSNs: 107.60 ± 12.39 ms, iMSNs: 93.67 ± 27.10 ms, Fig. 3a). This is also reflected in the peak lags obtained by cross-correlation analysis, indicating that membrane depolarization precedes whisker movement around the time of transition from quiescence to whisking (Supplementary Fig. 2). In both MSN types, membrane potentials were more depolarized during whisking than during quiescent epochs (dMSN control Q: −68.10 ± 1.40 mV, W: −64.99 ± 1.38 mV, *P* < 0.01; iMSN control Q: −69.84 ± 1.32 mV, W: −67.62 ± 1.67 mV, *P* < 0.01, Fig. 3a, c). In DA-depleted mice, the whisking-induced depolarization was abolished in dMSNs but not in iMSNs (dMSN 6-OHDA Q: −66.87 ± 1.90 mV, W: −66.31 ± 1.77 mV, *P* > 0.05; iMSN 6-OHDA Q: −66.68 ± 1.12 mV, W: −63.97 ± 1.21 mV, *P* < 0.01, Fig. 3b, c, d). This effect could also be seen in a cross-correlation analysis, where the magnitude of correlation between whisking and membrane potential was reduced in dMSNs in 6-OHDA mice (Supplementary Fig. 2). The membrane potential variance, measured as the standard deviation, was not significantly different between quiet and whisking epochs of the different groups, apart from iMSNs recorded in 6-OHDA mice (iMSN 6-OHDA Q: 2.02 ± 0.25 mV, W: 1.59 ± 0.17 mV, *P* < 0.05, Fig. 3b, d), however, the degree of depolarization was correlated with the intensity of whisking (Supplementary Fig. 3).

Our results show that MSNs in DLS encode and even predict whisking, and that DA depletion alters this coding in a pathway-dependent manner, reducing motor signals selectively in dMSNs.

## Sensory responses in MSNs are attenuated by whisking

Sensory-evoked responses represent a major source of synaptic input to MSNs in the DLS[16,23]. In different cortical regions, both enhancement[7,8,47] and attenuation of sensory responses by motor activity were observed[1,4,6]. Little is known about such sensorimotor interactions in the striatum. To examine whether and how whisking modulates sensory integration in MSNs, sensory stimuli were delivered as brief contralateral whisker deflections at random intervals (See Methods and Supplementary Movies 1 and 2). In parallel, whisker movements were recorded (Fig. 4a, green line), allowing for post hoc discrimination between stimuli delivered during periods of whisking or quiescence (Fig. 4a). Whisker deflections during quiescence evoked larger responses than those evoked during whisking in both dMSNs (dMSN control Q: 10.26 ± 3.09 mV; W: 5.32 ± 2.63 mV, *P* < 0.01, Fig. 4b, d) and iMSNs (iMSN control Q: 7.94 ± 1.71 mV; W: 4.35 ± 0.74 mV, *P* < 0.05, Fig. 4b, d). In contrast, no differences were observed in the latency of response peaks, neither in dMSNs (dMSN

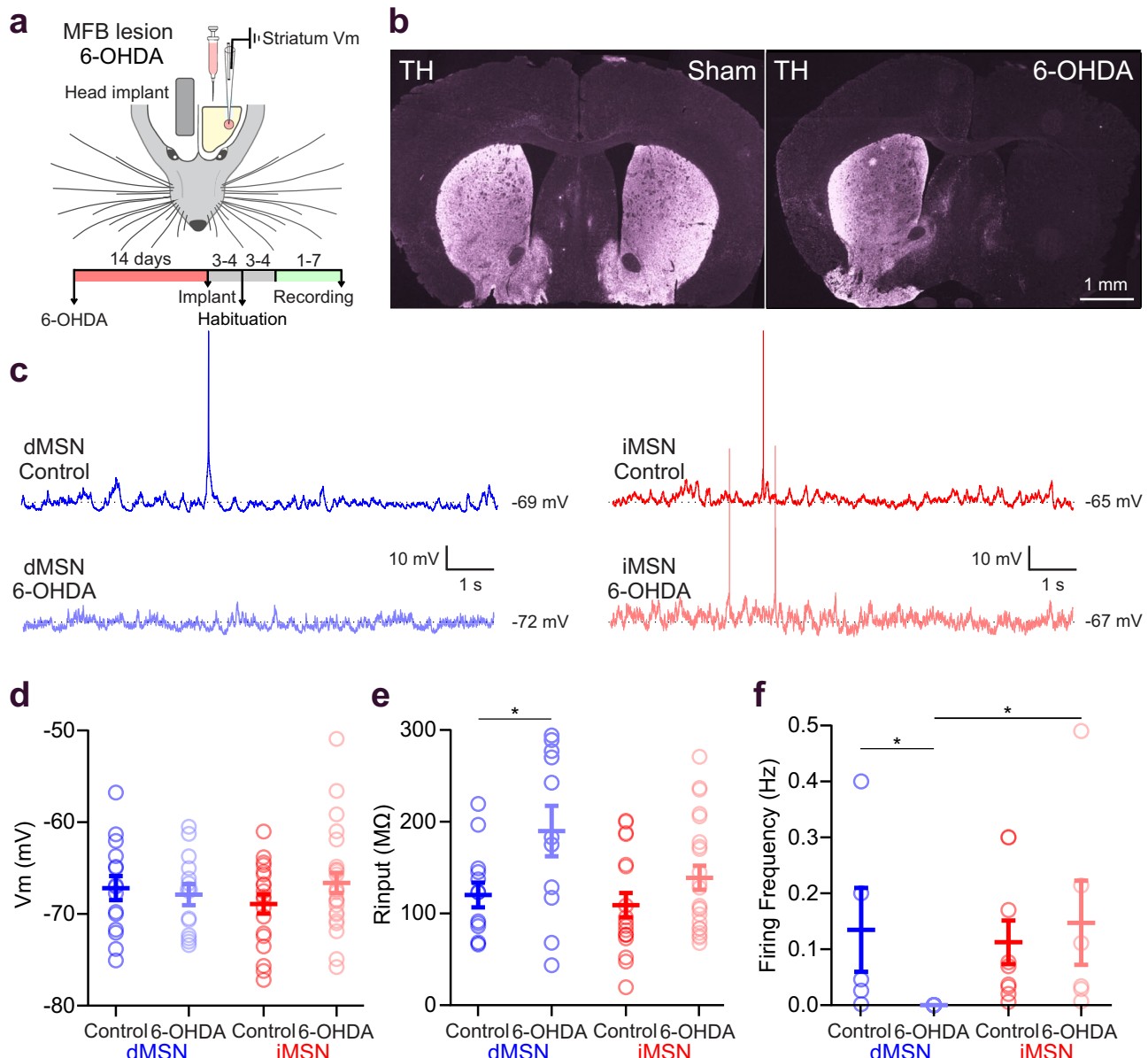

**Fig. 2 | Membrane properties of MSNs in control and 6-OHDA lesioned mice.**
**a** Schematic of the experimental setup for whole-cell recordings in dopamine-depleted mice and the timeline of the 6-OHDA lesion. **b** Coronal sections showing the striatal hemispheres of sham (left) and 6-OHDA lesioned (right) mice, stained for TH expression. Fluorescence was strongly reduced in the 6-OHDA lesioned hemisphere ($n = 12$ mice) compared to the control hemispheres. **c** Example traces showing spontaneous membrane potential activity in dMSNs (blue) and iMSNs (red) under control conditions (dark color) and in 6-OHDA lesioned mice (light color). **d** No differences were observed in the membrane potential (Vm) of dMSNs and iMSNs in control and 6-OHDA lesioned mice (dMSN control = −67.18 ± 1.30 mV, $n = 15$ cells; dMSN 6-OHDA = −67.89 ± 1.15 mV, $n = 14$ cells; iMSN control = −68.90 ± 1.05 mV, $n = 19$ cells; iMSN 6-OHDA = −66.62 ± 1.10 mV, $n = 25$ cells, $P > 0.05$ for all comparisons, one way ANOVA, Tukey multiple comparison test between groups, $F = 0.813$, $P = 0.491$). **e** The input resistances (R input) of dMSNs and iMSNs were similar in control conditions (dMSN control=120.10 ± 13.59 MΩ, $n = 13$ cells; iMSN control=114.49 ± 12.92 MΩ, $n = 17$ cells; $P = 0.995$). Following DA depletion, input resistance increased in dMSNs but not iMSNs (dMSN control vs. dMSN 6-OHDA =

189.96 ± 27.47 MΩ, $n = 11$ cells; * $P = 0.045$; iMSN control vs. iMSN 6-OHDA = 138.91 ± 13.13 MΩ, $n = 22$ cells; $P = 0.633$; dMSN 6-OHDA vs iMSN 6-OHDA, $P = 0.141$, one way ANOVA, Tukey multiple comparison test between groups $F = 3.582$, $P = 0.019$). **f** Spontaneous action potential discharge frequency of dMSNs and iMSNs showed no differences in control condition (dMSN control= 0.135 ± 0.074 Hz, $n = 5/15$ cells; iMSN control = 0.089 ± 0.035 Hz, $n = 8/19$ cells, two-sided Mann-Whitney test, $P = 0.943$). Following DA depletion, dMSNs did not exhibit spontaneous firing (0/14 cells) while iMSNs did not change their firing frequency (iMSN control vs iMSN 6-OHDA = 0.147 ± 0.075 Hz, $n = 6/25$ cells, two-sided Mann-Whitney test, $P = 0.754$). Spontaneous firing proportions for dMSN control vs dMSN 6-OHDA, two-sided Fisher's test, * $P = 0.042$; iMSN control vs iMSN 6-OHDA, two-sided Fisher's test, $P = 0.539$; dMSN 6-OHDA vs iMSN 6-OHDA, two-sided Fisher's test, * $P = 0.035$. Control dMSNs are in dark blue, control iMSNs are in dark red, 6-OHDA lesioned dMSNs are in light blue, and 6-OHDA lesioned iMSNs are in light red. For all panels, data are presented as mean ± SEM. Source data are provided as a Source Data file. MFB medial forebrain bundle.

control Q: 32.89 ± 2.10 ms; W: 33.90 ± 3.59 ms, $P > 0.05$, Fig. 4b, e) nor iMSNs (iMSN control Q: 30.90 ± 1.24 ms; W: 28.63 ± 2.08 ms, $P > 0.05$, Fig. 4b, e). Differences between the amplitude of sensory responses during quiescence and whisking were maintained also in DA-

depleted mice, in both dMSNs (dMSN 6-OHDA Q: 7.54 ± 1.42 mV; W: 4.62 ± 1.10 mV, $P < 0.01$, Fig. 4c, d) and iMSNs (iMSN 6-OHDA Q: 8.90 ± 1.77 mV; W: 5.05 ± 1.47 mV, $P < 0.001$, Fig. 4c, d). The temporal properties of sensory responses were not affected following DA

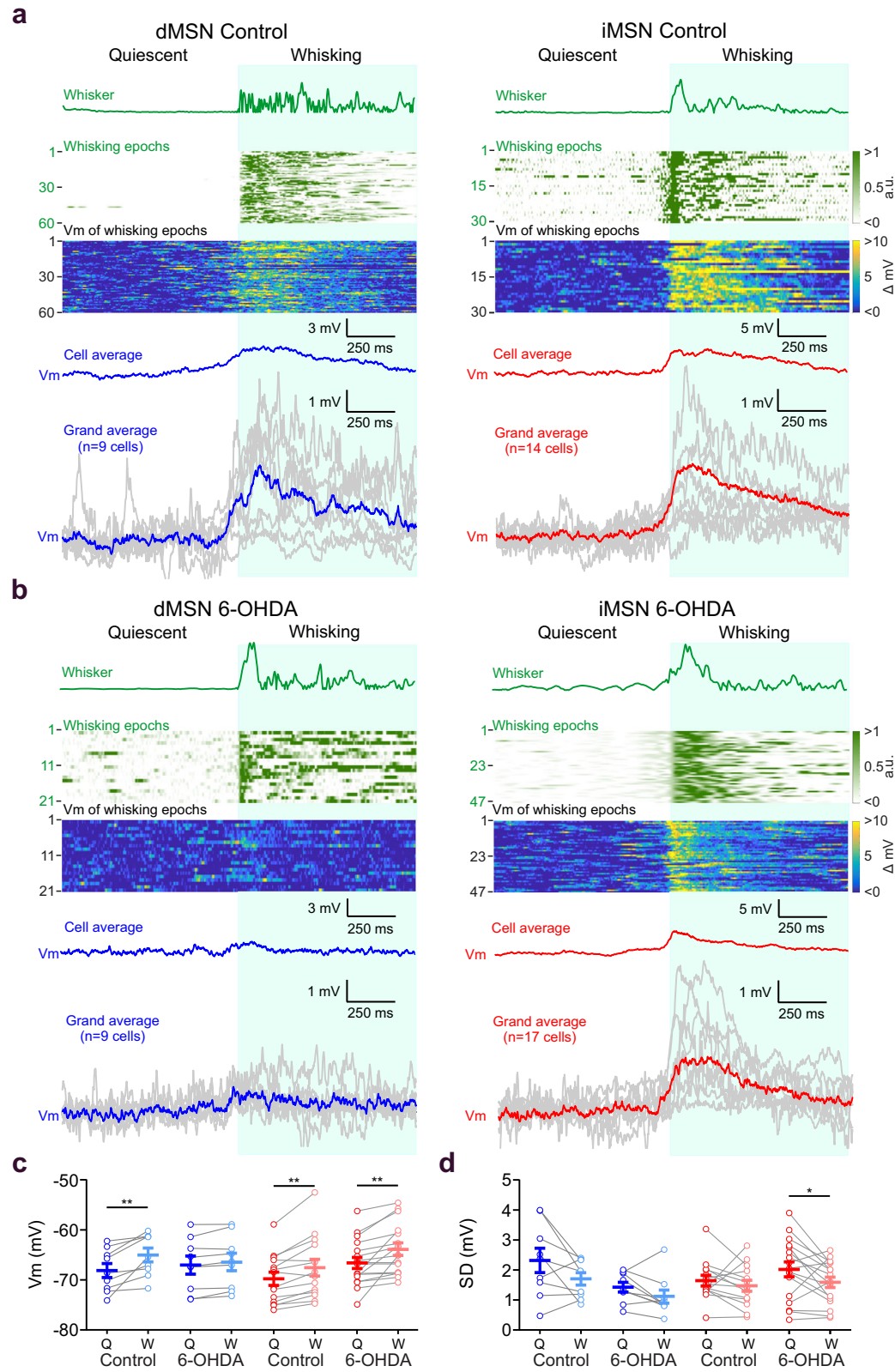

depletion, reaching peak depolarizations at similar times (Fig. 4e, $P > 0.05$). In a subset of cells, responses to ipsilateral stimulation were measured, also showing larger responses during quiescence than during whisking in both dMSNs and iMSNs (Supplementary Fig. 4).

The attenuation in sensory responses of MSNs could reflect changes in their membrane conductance during whisking. To test this,

we monitored the input resistance of responding MSNs during quiescence and whisking by injecting brief negative current steps (Fig. 5a). These measurements showed that the input resistance of MSNs was indeed lower during whisking than during quiescent periods (Fig. 5a–c). The attenuation in sensory responses of MSNs could also be caused by changes in cortical responses to whisker stimulation during quiescence and whisking[1]. Cortical LFP responses to whisker

**Fig. 3 | Whisker activity is preceded by membrane depolarization of MSNs in control conditions but not in dMSNs in the DA-depleted striatum. a** Membrane potential (Vm) depolarization of dMSN in control conditions precedes whisking. Left from top to bottom: Example of whisker activity transition from quiescence to whisking (whisker, green). Heat map showing 60 whisking epochs in a single recording of a dMSN in control conditions. Corresponding heat map showing membrane potential fluctuations in a single dMSN in control conditions during each whisking epoch described above (60 repetitions). The average membrane potential for the same neuron is shown below (cell average, blue). Note that the depolarization starts a few milliseconds before the whisking movement. Grand average of the dMSNs; each gray trace indicates independent dMSN average activity (n = 9 cells). Right: Same as in the left panel but for iMSNs in control mice. Heat maps (30 repetitions) and grand average (n = 14 cells). **b** Same as in **a** but for 6-OHDA lesioned mice. Heat maps (21 repetitions in dMSN and 47 in iMSN). Grand average (n = 9 cells in dMSNs and n = 17 in iMSNs). Note the absence of the depolarization during whisking in dMSNs of 6-OHDA lesioned mice in contrast with dMSNs under control conditions. **c** Membrane potential of dMSNs (blue) and iMSNs (red) during quiescent (Q) and whisking (W) epochs in control and 6-OHDA lesioned mice (dMSN control Q = −68.10 ± 1.40 mV; dMSN control W = −64.99 ± 1.38 mV, two-sided Wilcoxon paired test, ** P = 0.0039; dMSN 6-OHDA Q = −66.87 ± 1.90 mV; dMSN 6-OHDA W = −66.31 ± 1.77 mV, two-sided Wilcoxon paired test, P = 0.129; iMSN control Q = −69.84 ± 1.32 mV; iMSN control W = −67.62 ± 1.67 mV, two-sided paired t test, ** P = 0.001; iMSN 6-OHDA Q = −66.68 ± 1.12 mV; iMSN 6-OHDA W = −63.97 ± 1.21 mV, two-sided paired t test, ** P = 0.002). **d.** Standard deviation (SD) of the membrane potential of dMSNs and iMSNs during Q and W epochs (dMSN control Q = 2.32 ± 0.41 mV; dMSN control W = 1.70 ± 0.21 mV, two-sided Wilcoxon paired test, P = 0.156; dMSN 6-OHDA Q = 1.42 ± 0.16 mV; dMSN 6-OHDA W = 1.11 ± 0.22 mV, two-sided Wilcoxon paired test, P = 0.098; iMSN control Q = 1.65 ± 0.18 mV; iMSN control W = 1.47 ± 0.18 mV, two-sided paired t test, P = 0.374; iMSN 6-OHDA Q = 2.02 ± 0.25 mV; iMSN 6-OHDA W = 1.59 ± 0.17 mV, two-sided paired t test, * P = 0.047). For **c, d** each gray line represents the data from a single cell during Q (dark color) and W (light color), and error bars indicate mean ± SEM (blue). Source data are provided as a Source Data file. a.u. arbitrary units.

deflection were smaller when delivered during whisking (Fig. 5d–f), suggesting that sensory mediated activity was attenuated also at the presynaptic cortical level. These results show that responses to whisker deflection in MSNs are attenuated by ongoing self-generated whisking. This attenuation is mediated by both postsynaptic and presynaptic mechanisms in the corticostriatal pathway.

### DA depletion impairs bilateral discrimination in MSNs

We next examined how bilateral tactile stimuli are processed in DLS neurons, and whether bilateral responses are altered following DA depletion. To this end, we performed whole-cell recordings in awake mice while brief air puffs were delivered to the ipsi- or contralateral whiskers (Fig. 6a). We verified that unilateral air puffs did not induce any detectable movement in the opposite whiskers (Supplementary Movie 1). Ipsi- and contralateral whisker deflections delivered during quiescence evoked sensory responses in both dMSNs and iMSNs (Fig. 6b, d). In control mice, contralateral whisker stimulation evoked a larger short-latency depolarization (within 50 ms from the stimulus onset) than ipsilateral stimulation, in both dMSNs and iMSNs (P < 0.05, Fig. 6c, e, Supplementary Table 1). The onset delay of contralateral responses was also shorter than that of ipsilateral responses (P < 0.05, Fig. 6c, e, Supplementary Table 1), although no differences were seen in the time of the peak amplitude (P > 0.05, Fig. 6c, e, Supplementary Table 1). Following DA depletion, the differences between ipsi- and contralateral responses, both in terms of amplitude and onset latency, were abolished (P > 0.05, Fig. 6c, e, Supplementary Table 1). This effect of DA depletion was not apparent in cortical LFP recordings (Supplementary Fig. 5), suggesting that the observed attenuation occurs at the striatal level and may be caused by pathway specific alterations in corticostriatal and thalamostriatal synapses.

### DA depletion reduces the late sensory component in dMSNs

Sensory responses often had a secondary late component, between 100 and 250 ms after whisker deflection (Fig. 7). Interestingly, this late component was more pronounced when stimuli were delivered during whisker quiescence, while almost absent when stimuli were delivered during whisking (Supplementary Fig. 4). We tested whether the late component encodes stimulus laterality as the short-latency component (Fig. 6), and if it is affected by DA depletion. In control mice, ipsi- and contralateral whisker stimulation produced late responses with no differences in the amplitude, onset, or voltage time integral measured as area under the curve (AUC), in both dMSNs and iMSNs (P > 0.05, Fig. 7c, e, Supplementary Table 2). As in control mice, following DA depletion, no differences between ipsi- and contralateral responses were observed either in dMSNs or iMSNs (P > 0.05, Fig. 7c, e, Supplementary Table 2). However, the amplitude of the late response in

dMSNs was significantly smaller in DA-depleted mice (P < 0.05, Fig. 7c, Supplementary Table 2), which was not the case in iMSNs (P > 0.05, Fig. 7e, Supplementary Table 2). These results indicate that although not encoding for the laterality of whisker deflections, the late component is affected in a cell-type specific manner after DA depletion. One possible explanation for these differences could be the involvement of the late component in motor activity. Whisker deflection during quiescence often evoked whisking (QW trials, see also Supplementary Movie 2), however, in a smaller fraction of trials the air puff did not evoke a whisking bout (QQ trials, Supplementary Fig. 6a). We compared the late component in QQ vs. QW trials in control and DA-depleted mice. Whisker stimulation in control mice evoked larger late responses in QW than in QQ trials in both MSN types (Supplementary Fig. 6). Following DA depletion, iMSNs still had larger responses during QW than those recorded during QQ. In contrast, the late component in dMSNs was reduced, and no differences were observed between QQ and QW trials. Thus, the late component is correlated with stimulus-evoked whisking in both MSNs types, but was reduced only in dMSNs following DA depletion.

In summary, our results show that in the DLS, the same MSNs are engaged in both sensory integration and motor activity. Sensory responses are attenuated by whisking, and DA plays a role in the representation of both sensory and motor processes.

## Discussion

In this study we investigated the modulation of striatal sensory processing by motor activity and the changes resulting from DA depletion. We show that most neurons in the DLS encode both tactile sensory input (whisker stimulation) and self-generated movement (spontaneous whisking). Motor activity attenuated the sensory responses to tactile stimulation in both striatal MSN types, in both DA-intact and DA-depleted striatum. DA depletion reduced movement representation only in dMSNs, while impairing the bilateral sensory representation in MSNs of both pathways. Our study thus shows that motor activity shapes sensory responses in the striatum and that DA depletion compromises both sensory and motor representations in a cell type-dependent manner.

We showed that whisker movement was encoded by membrane potential depolarization in both dMSNs and iMSNs. Interestingly, depolarization preceded whisker movement onset, suggesting that the striatum is involved in preparatory neuronal activity occurring before movement initiation. In agreement with previous studies showing co-activation of dMSNs and iMSNs[48–52], our data suggest a cooperative rather than antagonistic role during action initiation. The depolarization of dMSNs upon whisker movement was abolished following DA depletion. This effect was cell type-dependent and not observed in iMSNs. The attenuation of the late component of sensory responses in

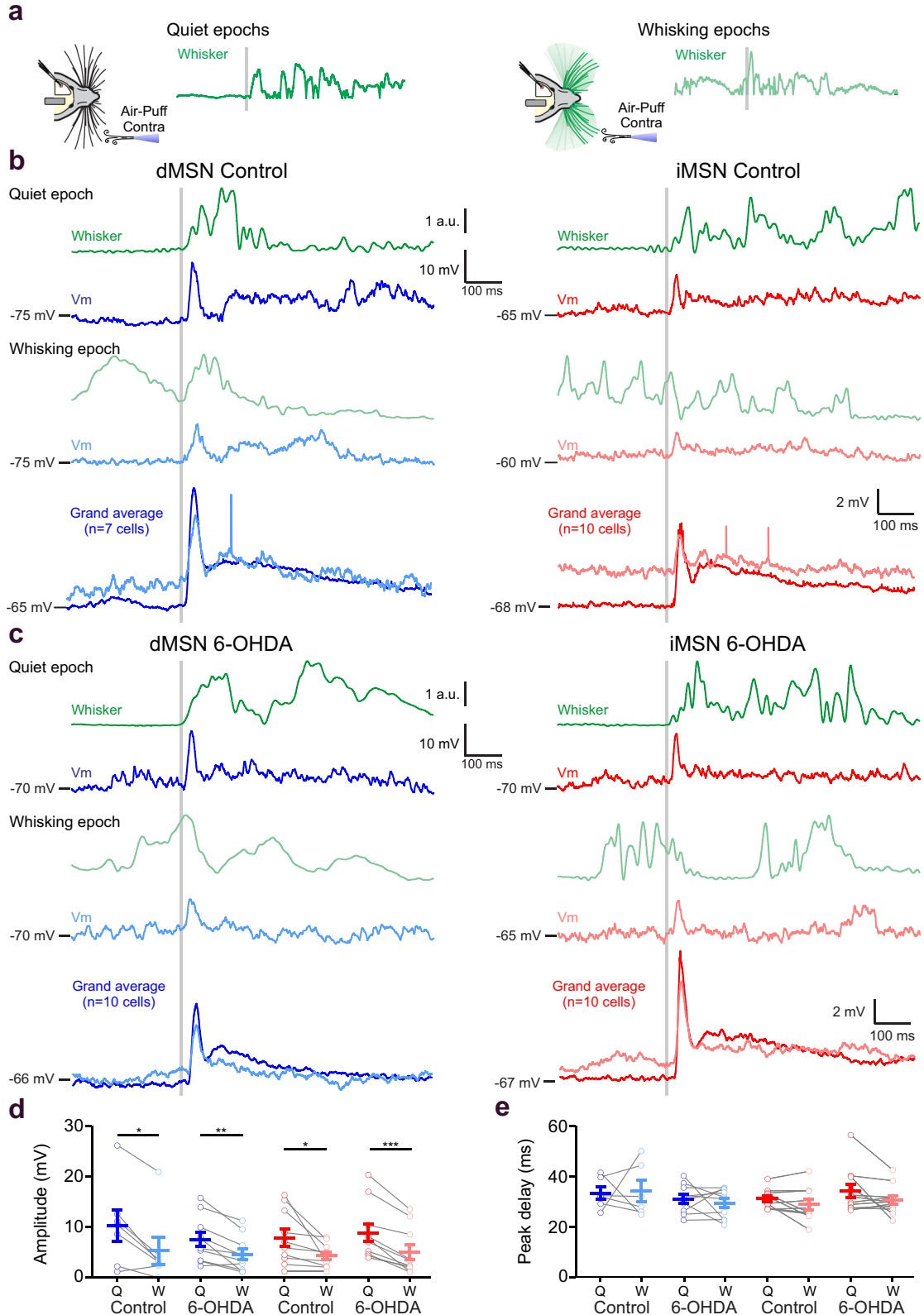

dMSNs (Fig. 7 and Supplementary Fig. 6) also points towards a cell type-specific effect of DA depletion. Together with the reduction in the spontaneous activity of dMSNs, our results show pathway-dependent changes in the striatal microcircuitry in PD, with reduction in the spontaneous and motor-related activity of the direct pathway. This effect of DA depletion may be due to the reduced excitability[53,54] or reduced corticostriatal input to dMSNs[23]. Supporting these results is a

recent study showing that diminished DA levels reduce the number of dMSNs recruited during locomotion[52]. Sensory-evoked movement was recently reported for auditory stimuli, which were shown to be effective in evoking stereotypic movements[55]. Interestingly, DA depletion did not significantly affect sensory-evoked whisking, which remained quite high, at ~90% success, in all cases (Supplementary Fig. 1). Moreover, there was little impact of the 6-OHDA lesion on whisking

**Fig. 4 | Sensory responses in MSNs are attenuated during whisking. a** Schematic of contralateral whisker stimulation during quiet epochs (left) or during whisking epochs (right). Stimuli were delivered randomly with intervals ranging from 3 to 6 s. Classification of events as occurring during quiescence (Q) or whisking (W) was done post-hoc. The gray line indicates the time point of the stimulus trigger. **b** Examples of traces from control dMSNs (blue left) and control iMSNs (red right) mice upon contralateral whisker stimulation (gray line). Dark color traces indicate the membrane potential (Vm) averaged response of individual MSNs to sensory stimulation during quiescence while light color traces indicate responses to sensory stimulation during whisking in the same cell. Whisker-related motor activity is indicated in green. Bottom panel, grand average of traces from independent MSNs in response to contralateral whisker stimulation in control mice (n = 7 cells in dMSNs and n = 10 cells in iMSNs). Note the decrease in amplitude in the sensory responses evoked during whisker movement in both MSN types. **c** Same as in **b** but for 6-OHDA lesioned mice. Dark color traces indicate MSNs responses to sensory stimulation during quiescence while light color indicate responses to sensory stimulation during whisking. Grand average (n = 10 cells in dMSNs and n = 10 cells in

iMSNs). **d, e** Contralateral whisker deflections during quiescence produce larger amplitude responses than during whisking in both MSN types in control and 6-OHDA lesioned mice **d** (dMSN control Q = 10.26 ± 3.09 mV; dMSN control W = 5.32 ± 2.63 mV, two-sided Wilcoxon paired test, * P = 0.031; dMSN 6-OHDA Q = 7.54 ± 1.42 mV; dMSN 6-OHDA W = 4.62 ± 1.10 mV, two-sided paired t test, ** P = 0.002; iMSN control Q = 7.94 ± 1.71 mV; iMSN control W = 4.35 ± 0.74 mV, two-sided paired t test, * P = 0.031; iMSN 6-OHDA Q = 8.90 ± 1.77 mV; iMSN 6-OHDA W = 5.05 ± 1.47 mV, two-sided paired t test, *** P < 0.001) without affecting the peak of maximum depolarization (**e**, peak delay) (dMSN control Q = 32.89 ± 2.10 ms; dMSN control W = 33.90 ± 3.59 ms, two-sided Wilcoxon paired test, P > 0.99; dMSN 6-OHDA Q = 30.62 ± 1.76 ms; dMSN 6-OHDA W = 28.98 ± 1.83 ms, two-sided paired t test, P = 0.500; iMSN control Q = 30.90 ± 1.24 ms; iMSN control W = 28.63 ± 2.08 ms, two-sided paired t test, P = 0.14; iMSN 6-OHDA Q = 34.20 ± 2.90 mV; iMSN 6-OHDA W = 30.21 ± 1.88 mV, two-sided paired t test, *** P < 0.11). For **d, e** each gray line represents the data from a single cell during Q (dark color) and W (light color), and error bars indicate mean ± SEM. Source data are provided as a Source Data file. a.u. arbitrary units.

behavior (Supplementary Fig. 1), suggesting that striatal dopamine is not essential for this behavior, in line with a recent study using a different PD model[56]. Another possibility is that compensatory mechanisms restore and maintain whisking despite dopamine depletion. It would be interesting to quantify whisking activity in different PD models and time points in future studies.

We show that both MSN types respond to bilateral whisker stimulation in awake mice. These results are in agreement with whole-cell recordings in anesthetized mice[16,21,23]. A previous study using whole-cell recordings in awake mice reported that sensory responses in dMSNs were markedly larger than in iMSNs[20], which was not apparent in our recordings. This discrepancy could be explained by differences in the experimental protocols, where only one whisker was stimulated. Here, we delivered the sensory stimuli as an air-puff to several whiskers. Moreover, the smaller responses in iMSNs compared to dMSNs may reflect the learned task in the aforementioned study[20]. Previous studies report enhancement of visual responses during movement and activated cortical states[2,7–9,47,57] and attenuation of auditory responses[4–6]. Here we showed that striatal responses to whisker deflection are attenuated during whisking. This attenuation of sensory responses during active periods can be explained by a decrease in excitatory synaptic transmission due to adaptation in presynaptic structures during whisking[1,58–60]. Another mechanism could be an increase in MSN membrane conductance due to ongoing synaptic input during whisking, and lastly, a decreased synaptic driving force due to membrane depolarization in MSNs. For instance, during up-states in anesthetized mice, a brain state characterized by depolarization and elevated synaptic input, sensory stimulation generates smaller responses in the MSNs than those evoked in the down-state[16,21]. Our data show that during whisking, MSNs undergo depolarization and increase in membrane conductance (Figs. 3 and 5), both supporting the attenuation of synaptic responses. The conductance increase is likely to be mediated by both afferent and local synaptic inputs, such as GABAergic connections from neighboring MSNs and interneurons[61]. In addition, we observed an attenuation at the presynaptic level, reflected in the reduced amplitude of the cortical LFP response during whisking. These results suggest that there are multiple presynaptic and postsynaptic mechanisms supporting the attenuation.

The early component of sensory responses (within 50 ms from the stimulus onset) has been linked to the initial whisker deflection[16,20,22,23]. It also encodes information associated with laterality of whisker stimulation in anesthetized mice[23]. We observed that MSNs responded to contralateral tactile inputs with earlier and larger depolarizing responses than to ipsilateral inputs. Whereas lateral encoding by MSNs is still present in awake mice, sensory responses were smaller and earlier than in anesthetized mice. Similar results were observed in the visual cortex, when comparing responses between anesthetized and

awake mice[62]. Following DA depletion, the bilateral asymmetry in the responses was abolished in both MSN types, rendering contralateral and ipsilateral sensory responses almost identical in terms of amplitude and latency. Cortical activity recorded as local field potentials in barrel cortex showed no changes in bilateral response asymmetry after DA depletion (Supplementary Fig. 5), suggesting that the loss of lateral encoding in MSNs is not caused by changes in the global cortical activity. What mechanisms could explain such effects of DA depletion? One possibility is that they are caused by differential changes in the synaptic properties of ipsilateral and contralateral striatal afferents. For example, DA depletion was shown to attenuate responses in MSNs to ipsilateral stimulation of the motor cortex[36,63] and to induce changes in thalamostriatal inputs[64,65]. Another possibility is that DA depletion has different impact on the intratelencephalic (IT) and pyramidal tract (PT) pathways[66], thus differentially affecting ipsi- and contralateral corticostriatal inputs as well as cortico-callosal projections[67]. The changes in the respective pathways could be directly on the axons of cortical projection neurons but could also be mediated indirectly via intrastriatal mechanisms such as cholinergic interneurons that were shown to be activated primarily by PT inputs[24,68] and regulate corticostriatal transmission[69]. The mechanisms underlying the changes in laterality coding are, however, not completely clear and should be further investigated in future studies.

The late component of the sensory response has been previously shown in both anesthetized and awake mice[16,20,23]. Our data showed that ipsi- and contralateral whisker stimulation produced similar late responses in terms of amplitude and latency, suggesting no encoding of stimulus laterality. Moreover, we show that the late component was larger in cases where whisker deflection triggered whisking (Supplementary Fig. 6). Following DA depletion, the amplitude of the late component was reduced in dMSNs but not in iMSNs. In addition, dMSNs in DA-depleted mice showed no difference between trials that triggered whisking and those that did not. These results, and the observed decrease in whisking representation in dMSNs after DA depletion (Fig. 3b), together suggest that the late component encodes sensory-evoked movement and not only sensory information per se.

In conclusion, we show that most neurons in the mouse DLS encode both sensory and motor processes, and that sensory responses are modulated by motor activity in both dMSNs and iMSNs. We also show that both sensory and motor representations are altered in the DA-depleted striatum, with motor processes being more affected in the direct pathway.

## Methods
### Experimental model and subject details
All experiments were performed according to the guidelines of the Stockholm municipal committee for animal experiments under an

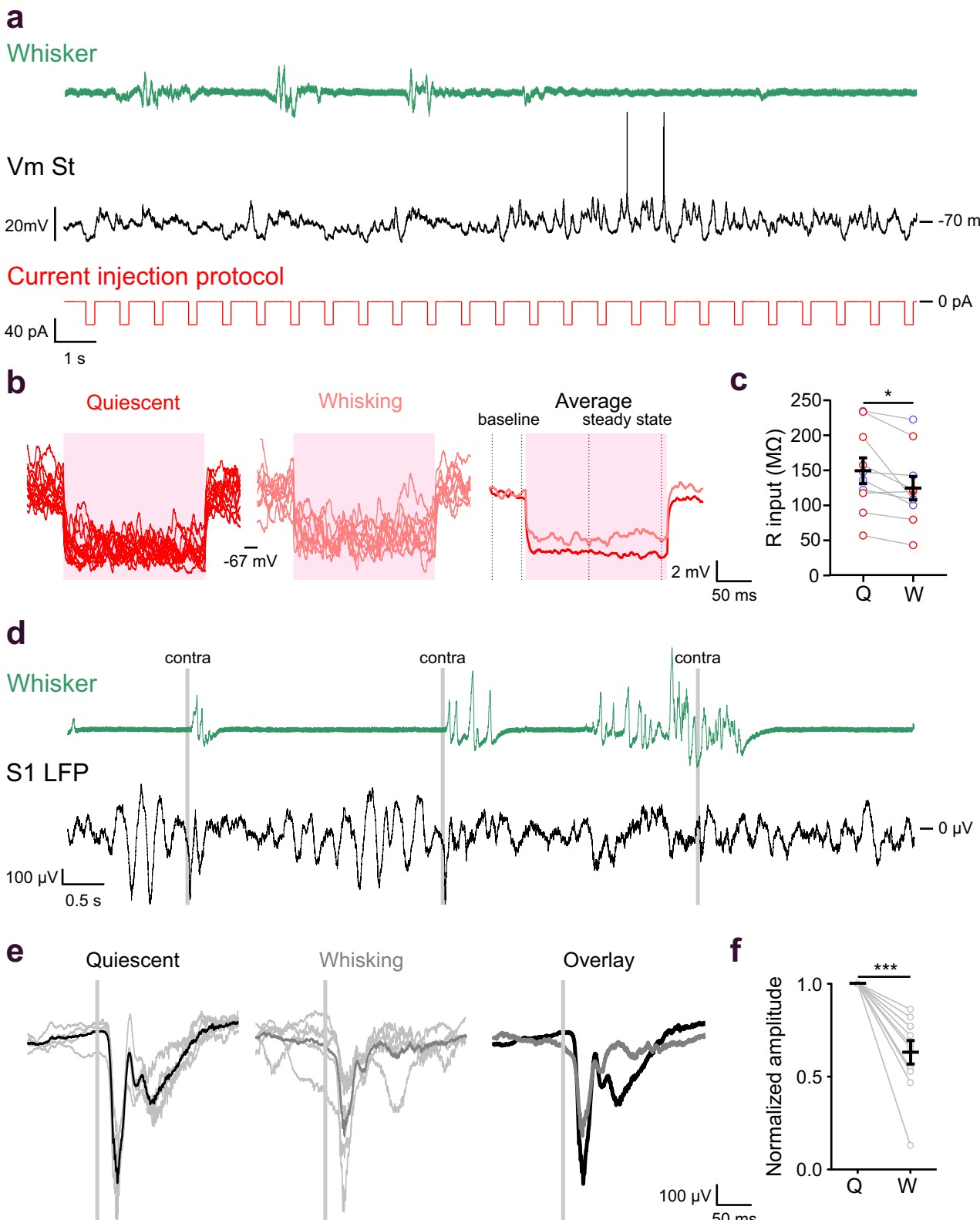

ethical permit to G.S. (N2022-2020). Mice ($N = 82$) of both sexes between 2 and 3 months of age were housed in groups from two to five in polycarbonate individually ventilated cages equipped with environmental enrichment (carton tubes and wood sticks). The cages were kept under constant temperature and humidity with a 12 h light/dark cycle and *ad libitum* access to food and water. D1-Cre (B6.FVB(Cg)-Tg(Drd1-cre)EY217Gsat/Mmucd, GENSAT, MMRRC_034258-UCD), D2-

Cre (STOCK Tg(Drd2-cre)ER44Gsat/Mmucd, GENSAT, MMRRC_017263-UCD) or Adora2a-Cre (STOCK Tg(Adora2a-cre)KG139Gsat/Mmucd, GENSAT, MMRRC_031168-UCD) mice were crossed with the Channelrhodopsin (ChR2)-YFP reporter mouse lines (B6;129S-Gt(ROSA)26Sortm32(CAG-COP4*H134R/EYFP)Hze/J, the Jackson Laboratory, IMSR_JAX:012569) to induce expression of ChR2 in either dMSNs or iMSNs.

**Fig. 5 | Mechanisms supporting the attenuation of the response to whisker stimulation delivered during whisking compared to quiescence. a** Example trace showing spontaneous membrane potential activity as recorded from an iMSN in the DLS of a spontaneously whisking mouse. Membrane potential (Vm, black) of the neuron was recorded simultaneously with the measurement of whisker activity (Whisker, green) while injecting hyperpolarizing current steps (−40 pA, 200 ms, red). **b** The responses to current injections occurring during quiescence (red traces, left) and whisking (light red traces, right) were averaged separately. Input resistance was calculated using the difference between the average membrane potential 100 ms before the current pulse (baseline) and the average membrane potential of the last 100 ms of the current pulse (steady state). **c** Input resistance during whisking (W) was significantly lower than during quiescence (Q) epochs (Q = 148.77 ± 18.50 MΩ; W = 123.99 ± 16.74 MΩ, $n$ = 10 cells, two-sided paired t test, *

$P$ = 0.02). **d** Example trace showing LFP activity recorded in primary somatosensory cortex during whisker stimulation. Cortical activity (LFP S1, black) was recorded simultaneously with the measurement of whisker activity (Whisker, green), during random whisker stimulations (trigger indicated as light gray lines). **e** Cortical responses to whisker stimulation delivered during quiescence (black, left) and spontaneous whisking (gray, middle). Responses are presented as single repetitions (light gray) overlaid with an average trace as indicated. The average traces for the two conditions are presented as an overlay (right). **f** Amplitude of the cortical responses to whisker stimulation delivered during whisking (W) is significantly lower that during quiescence (Q) epochs (normalized responses; Q = 1; W = 0.63 ± 0.06, $n$ = 11 mice, two-sided paired t test, *** $P$ < 0.001). Data are presented as mean value ± SEM. Source data are provided as a Source Data file. St striatum, LFP local field potential.

## 6-OHDA lesion

Adult mice of both sexes between 2 and 3 months of age were anesthetized with isofluorane (AbbVie AB, Cat: 506949) and mounted in a stereotaxic frame (Stoelting). Mice received one unilateral injection of 1 μL of 6-OHDA-HCl (3.75 μg/μL dissolved in 0.02% ascorbic acid, Sigma-Aldrich, CAS: 28094-15-7) into the medial forebrain bundle at the following coordinates (in mm): antero-posterior −1.2, medio-lateral +1.2, and dorso-ventral −4.8. After surgery, all mice were injected with Temgesic (0.1 mg/Kg, Indivior Europe Limited (Apoteket), Cat: 521634) and allowed to recover for at least 2 weeks. Mice, injected with 6-OHDA that showed rotational behavior were used in our experiments[70].

## Head implants

After a recovery period from previous treatments, mice were anesthetized with isofluorane, and the head was fixed in a stereotaxic apparatus. Temgesic (0.1 mg/Kg) was administered before surgery. The body temperature was maintained at 36.5/37 °C by a heating pad. An ocular ointment (Viscotears 2 mg/g, Alcon (Apoteket), Cat: 529807) was applied over the eyes to prevent ocular dehydration. Lidocaine (AstraZeneca, Cat: 236984) was applied on the skin surface before the incision. The skin covering the regions of interest was removed, and the bone gently cleaned. Targeted regions for intracellular and LFP recordings were marked using stereotaxic coordinates on the surface of the skull. Somatosensory cortex (S1) craniotomy coordinates for LFP recordings: antero-posterior −1.5 mm to bregma, +3.2 mm medio-lateral to midsagittal suture. Striatum craniotomy coordinates for intracellular recordings: antero-posterior 0 mm, medio-lateral +3 mm. Then, a thin layer of light-curing adhesive (Ivoclar Vivadent, Cat: 665156WW) was applied on the exposed skull. An aluminum metal head-post was fixed with dental cement Tetric Evo (Ivoclar Vivadent, Cat: 595953WW) to the right hemisphere. A wall of dental cement was built along the edge of the bone covering the left hemisphere. After the surgery, the animals were returned to their home cage. Three days after the implantation, mice were habituated to being head-restrained over a period of 3-4 days. The recordings were performed up to 7 days after the habituation.

## In vivo field and whole-cell recordings

On the day of the experiments, mice were anesthetized with isoflurane (3% for anesthesia induction then 1.5–2%), and small craniotomies (300–500 μm in diameter) were drilled to access the targeted areas. The open craniotomies were covered with silicone sealant (Kwik-Cast, WPI, Cat: MSPP-KWIKCAST), and the animals were returned to their home cages for recovery. After 2–4 h of recovery, mice were head-fixed, and the silicone from the craniotomies was removed. The body was restrained in a tube, preventing mice from performing large trunk and limb movements. A bipolar tungsten electrode with an impedance of 1–2 MΩ was inserted 1 mm deep from the surface at the Barrel Field craniotomy (1.5 mm A-P; 3.25 mm M-L). Signals were acquired using a Differential AC Amplifier model 1700

(A-M Systems, USA) and digitized at 20 KHz with CED and Spike2 (Cambridge Electronic Design) parallel to whole-cell recordings. Whole-cell patch-clamp recordings were performed in the DLS (0 mm A-P; 3 mm M-L) from 2 to 2.5 mm below the pia, a region that receives projections from sensory and motor areas[71]. Patch pipettes were pulled with a puller P-1000 (Sutter Instruments). Pipettes (7–10 MOhm, borosilicate, Sutter, Cat: GBF150-86-7.5HP) contained (in mM): 130 K-gluconate, 5 KCl, 10 HEPES, 4 Mg-ATP, 0.3 GTP, 10 Na2-phosphocreatine, and 0.2–0.3% biocytin (pH = 7.2–7.3, osmolarity 280–290 mOsm). Signals were amplified using MultiClamp 700B amplifier (Molecular Devices) and digitized at 20 KHz with a CED acquisition board and Spike2 software (Cambridge Electronic Design). All patch-clamp recordings were obtained in current-clamp mode without current injection. For characterization of intrinsic electrophysiological properties, current injections from −110 pA to +110 pA in steps of 20 pA for 3 s each were applied. Values for quiet and whisking states for each current injection were extracted separately. Membrane potential values were not corrected for liquid junction potential. The optopatcher was used for the online identification of recorded MSNs[46] (A-M systems, WA USA). Light steps of 500 ms were delivered at maximum LED power (3 mW at the tip of the fiber, 470 nm, Mightex systems) through an optic fiber inserted into the patch-pipette while recording the responses in whole-cell configuration. Positive cells responded to light pulses by a step-like depolarization, often exhibiting AP discharge, while negative cells did not show any response to light pulses.

## Cell labeling and immunohistochemistry

The cells were loaded with biocytin (Sigma, Cat: B4261-100MG) during the recording (>5 min recordings). At the end of the recording session, mice were sacrificed with an overdose of sodium pentobarbital (200 mg/kg I.P.) and transcardially perfused with 4% PFA in 0.01 M PBS. The brain was extracted and kept for an additional 2 h treatment in PFA, after which it was transferred to 0.01 M PBS. Before 24 h of sectioning, the brain was transferred to and kept in 12% sucrose solution in 0.01 M PBS. Coronal cryo-sections of 20 μm were mounted on microscope gelatin-coated slides and incubated overnight with Cy3 conjugated streptavidin (1:1000, Jackson ImmunoResearch Laboratories, Cat: 016-160-084) in staining solution (1% BSA, 0.1% Na Deoxycholate, and 0.3% triton in 0.01 PBS) at 4 °C. After washing in PBS, slides were mounted, and images were acquired on a fluorescence microscope in order to locate the recorded cell. In YFP positive mice, Cy3 positive cells were examined for YFP signal for classification as D1 or D2 expressing MSNs.

## TH Immunofluorescence

For TH immunofluorescence, coronal cryo-sections (20 μM) from the striatum were collected, mounted, rinsed in PBS and incubated in blocking solution (5% normal donkey serum, 0.3% Triton X-100, 1% BSA in PBS) for 30 min. Sections were incubated in rabbit anti-TH polyclonal antibody (Millipore, Cat: AB152) diluted 1:1000 in 1% PBS-BSA at

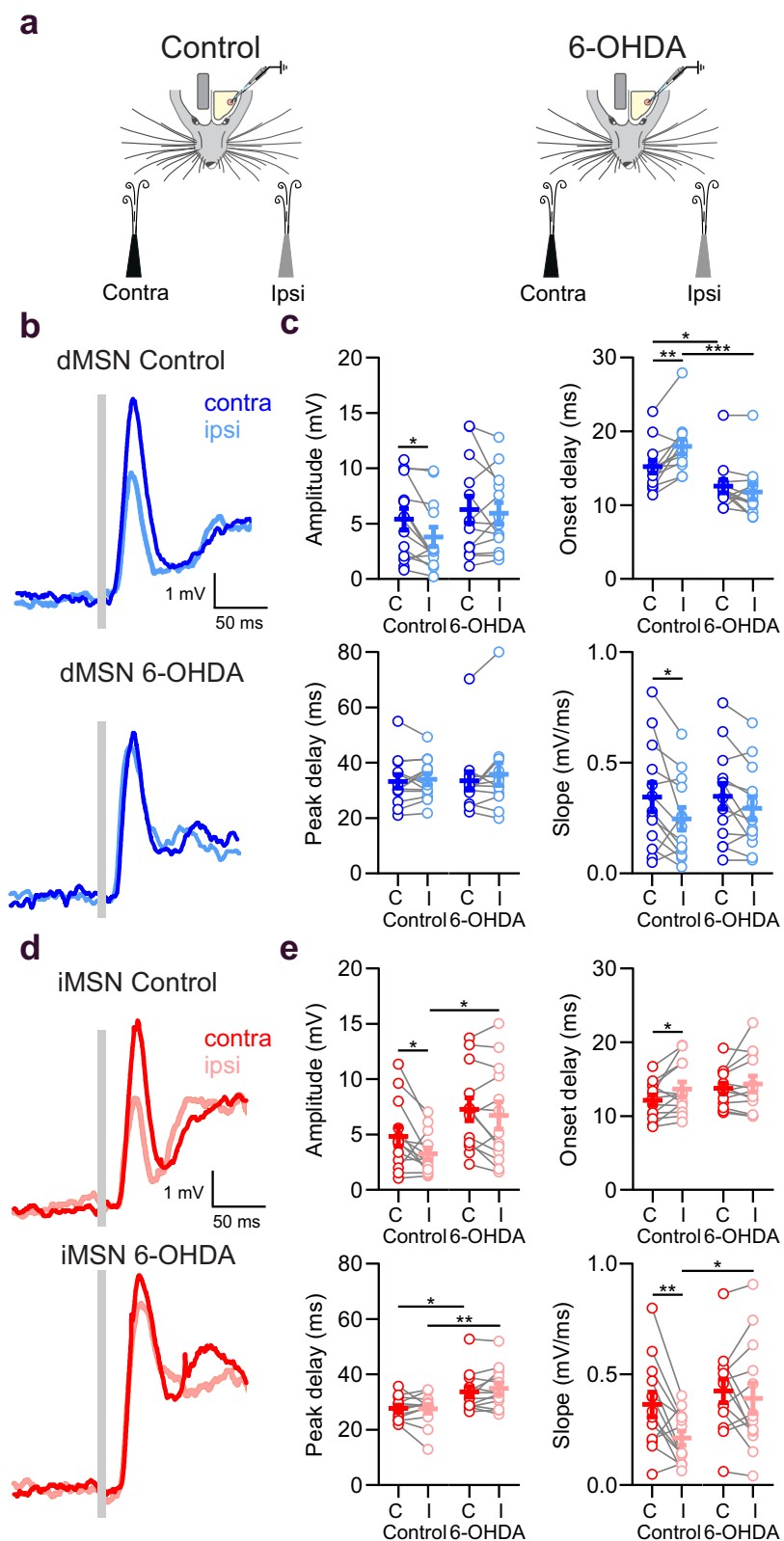

4 °C overnight. The sections were rinsed in PBS and incubated for 2 h at room temperature in Cy3-conjugated goat anti-rabbit polyclonal secondary antibody (Jackson Laboratories, Cat: 111-165-003) diluted 1:500 in 1% BSA-PBS. After processing, sections were examined with Zeiss Axio Imager M1 microscope (Carl Zeiss) and images of the dorsal striatum were captured at X4 magnification. TH staining showed over 80% reduction in fluorescence compared to unlesioned hemispheres.

**Whisking detection via reflector sensor**

During whole-cell recordings, whisking epochs were detected by IR reflective sensor (HOA1405, 574 Honeywell, NC, USA)[67]. It was positioned pointing toward the whisker pad at a distance of 5 mm. The signal was acquired at 20 KHz using Spike2 software. In order to identify the whisking epochs, 2 s of quiescence whisker signal was computed and amplitudes larger than 1.5 times the signal's STD was

**Fig. 6 | Laterality is encoded in the early component of sensory responses and impaired following DA depletion. a** Schematic of contralateral (contra) and ipsilateral (ipsi) whisker deflections during quiet epochs in control and 6-OHDA lesioned mice. The stimulations were delivered randomly in intervals from 3 to 6 s. **b** Grand average of responses of dMSNs in control (top) and 6-OHDA (bottom) lesioned mice to contralateral (dark blue traces) and ipsilateral (light blue traces) whisker stimulation. The gray bar indicates the moment the stimulation was delivered. **c** Differences in response amplitude (dMSN control C = 5.55 ± 0.98 mV; dMSN control I = 3.95 ± 0.88 mV, two-sided paired t test, * P = 0.01), onset delay (dMSN control C = 15.20 ± 0.87 ms; dMSN control I = 17.94 ± 0.99 ms, two-sided paired t test, ** P = 0.002; dMSN control C vs dMSN 6-OHDA C = 12.54 ± 0.89 ms; two-sided unpaired t test, * P = 0.04; dMSN control I vs dMSN 6-OHDA I = 11.77 ± 1.01 ms; two-sided unpaired t test, *** P < 0.001), and slope (dMSN control C = 0.34 ± 0.07 mV/ms; dMSN control I = 0.25 ± 0.05 mV/ms, two-sided paired t test, * P = 0.03) of dMSNs between contralateral (C), and ipsilateral (I) responses

(n = 13) are abolished in 6-OHDA lesioned mice (n = 13). **d** Grand average of responses of iMSNs in control (top) and 6-OHDA (bottom) lesioned mice to contralateral (red) and ipsilateral (light red) whisker stimulation. **e** Differences in response amplitude (iMSN control C = 4.84 ± 0.87 mV; iMSN control I = 3.27 ± 0.52 mV, two-sided paired t test, * P = 0.03, iMSN control I vs iMSN 6-OHDA I = 6.71 ± 1.22 mV, two-sided unpaired t test, * P = 0.02), onset delay (iMSN control C = 12.14 ± 0.69 ms; iMSN control I = 13.64 ± 0.98 ms, two-sided paired t test, * P = 0.03), and slope (iMSN control C = 0.36 ± 0.06 mV/ms; iMSN control I = 0.21 ± 0.03 mV/ms, two-sided paired t test, ** P = 0.006; iMSN control I vs iMSN 6-OHDA I = 0.39 ± 0.07 mV/ms, two-sided unpaired t test, * P = 0.03) of iMSNs between contralateral (C), and ipsilateral (I) responses (n = 13) are abolished in 6-OHDA lesioned mice (n = 13). For **c**, **e** each gray line represents the data from a single cell and error bars indicate mean ± SEM. Complete summary of values is shown in Supplementary Table 1. Source data are provided as a Source Data file.

---

defined as a whisking epoch. Threshold crossing epochs with inter-epoch-interval smaller than 200 ms were binned together and defined as whisking epochs. Whisking frequency was inferred from the total number of positive peaks in 1 s interval after whisking onset.

### Change in spontaneous activity in the quiet to whisking transition
Epochs including the transition from quiescence to whisking activity were selected, and one second time window was used to calculate the parameters of spontaneous activity for each behavior. After removing action potentials using a median filter, at least 10 epochs were considered to calculate the mean membrane potential and its variance. Heat maps were created with the function 'imagesc' in Matlab 2020a. All recordings were aligned to the onset of the movement.

### Whisker stimulation
Air puffs were delivered by picospritzer positioned 3 centimeters from the mouse's whiskers edge. Randomized 15 ms air puff stimulations (ipsi- and contralateral) were delivered at a range of 0.2 to 0.33 Hz at least 10 responses were acquired for each stimulation condition. The air pressure was set to 15 psi. Air puffs were aimed towards the distal parts of the whiskers and did not evoke movements in the opposite whisker (Supplementary Movie 1) or cause an aversive response, seen as an eye-blink (Supplementary Movie 2). For the analysis, sensory responses evoked by air-puff stimulation were first sorted according to ipsi- or contralateral origin. Subsequently, these two groups were sorted according to the mouse behavior and then averaged to calculate the responses properties (amplitude, onset delay, peak delay, and slope) for quiescence or whisking.

### Video recording of whisker activity
To analyze the impact of ipsilateral whisker stimulation on the contraleral whisker pad and vice versa, we used a high speed video camera located high up allowing to distinguish the different vibrissae recording at 100 FPS (Grasshopper3, GS3-U3-23S6M-C, FLIR Systems, Wilsonville) using FlyCapture 2.13.3.31 (FLIR Systems, Wilsonville). To avoid the interference of spontaneous whisking, mice were anaesthetized by intraperitoneal (IP) injection of ketamine (75 mg/kg) and medetomidine (1 mg/kg) diluted in 0.9% NaCl and air puffs stimulations to the contralateral and ipsilateral whisker pads alternately were delivered as previously described. To track whisker movements, we used DeepLabCut 2.2[72,73]. To train DeepLabCut, first two labels were placed in an ipsilateral (red and yellow marks), and a contralateral whiskers (blue and purple) in 50 frames chosen manually from the video. Once the network was trained, the videos were analyzed to extract the trajectories of the whiskers. In the second video, we measured the area of the eye in awake mice to demonstrate the lack of aversion after whisker stimulation. In the recorded video,

we labeled four points in the eye forming a trapezoid and as in the previous videos, we manually marked 50 frames in the recorded video to train the network and infer the trajectories. The eye size is calculated as the area that is form after connecting the four labels in the eye using the functions 'polyshape' and 'area' from Matlab R2020a. To improve visualization, the videos are played at 30 FPS (Supplementary Movies 1-2).

### Membrane potential and whisker activity correlations
The Pearson correlation between the membrane potential and the whisking signal was calculated for each neuron using 2 s windows containing 1 s prior and 1 s after whisking onset. We used the 'xcorr' function in Matlab R2020a with the setting normalization to 'coeff'. The whisking epochs were identified as described previously in the section 'Whisking detection via reflector sensor'. Signals were low-pass filtered below 10 Hz using a Butterworth filter before extracting the correlation coefficient. The cross-correlation for each repetition was average, giving a single value per neuron. Correlation values and lag time correspond to the maximum correlation, and the time it was obtained.

### Characteristics of spontaneous and evoked whisker activity
The spontaneous whisking behavior of control and 6-OHDA lesion mice was analyzed from at least 100 s of whisker activity recording. All whisking epochs were extracted as described in the section 'Whisking detection via reflector sensor for each animal. The percentage of time whisking was calculated as the proportion of time whisking with respect to the total recording time.

For the probability histogram, all the whisking epochs obtained for both groups (control n = 416 bouts from 14 mice; 6-OHDA n = 403 bouts from 14 mice) were distributed in 100 ms bins from 0 to 12 s corresponding to the most extended whisking epoch observed in the data. Bouts per minute were computed for each animal as the total number of whisking epochs divided by the total length of the recording in minutes. Mean, median, maximum, and minimum bout duration were obtained using the Matlab R2020a functions 'mean', 'median', 'max', and 'min', respectively, for all the whisking bouts for each mouse. For the evoked whisking activity, the percentage of contra- and ipsilateral whisker deflections that evoke whisker movement was analyzed and represented as a percentage of success.

### Quantification and statistical analysis
All data are represented as mean ± SEM. All data distribution was first checked for normality (Shapiro-Wilk test) and analyzed accordingly. Normally distributed data were tested by one-way ANOVA followed by post hoc Tukey's test analysis for multiple comparisons, and the unpaired and paired two-sample Student's t-test was used for two-group comparisons. Non-normally

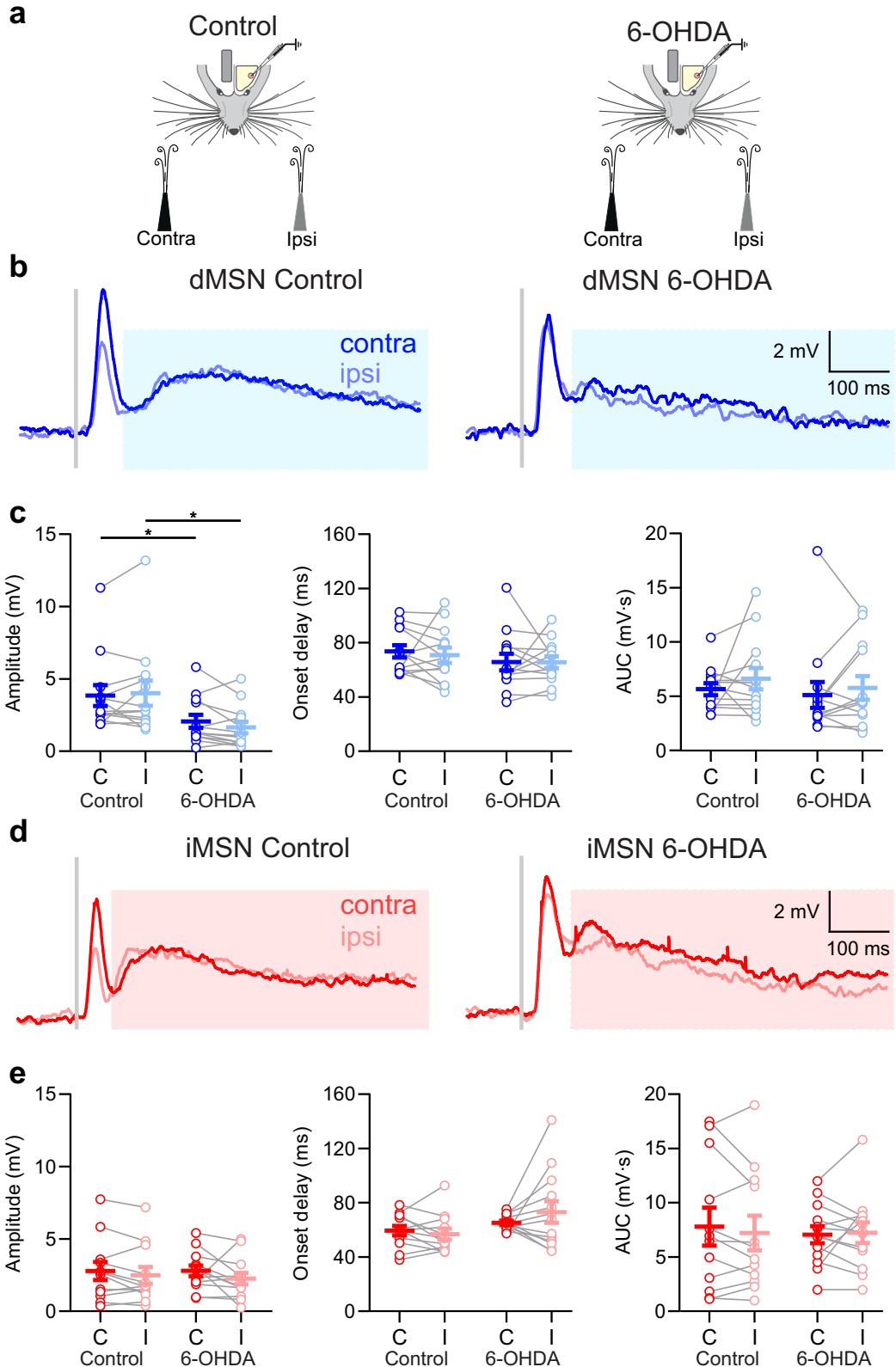

distributed data were analyzed by the Kruskal-Wallis test for multi-group comparisons, followed by Mann-Whitney for two-group comparison, Wilcoxon signed-rank test was used for paired samples. Spontaneous firing proportions were analyzed using Fisher's exact test. Statistical analyses were done in Prism 8 and Matlab R2020a. Final figures have been assembled with Corel DrawX8.

**Reporting summary**

Further information on research design is available in the Nature Portfolio Reporting Summary linked to this article.

## Data availability

A source data file supporting the findings of this study is provided with this paper and its supplementary information files. Further

**Fig. 7 | The late component of the sensory response is not involved in laterality encoding and is reduced in dMSNs following DA depletion. a** Schematic of contralateral (contra) and ipsilateral (ipsi) whisker deflections during quiet epochs in control and 6-OHDA lesioned mice. The stimulations were delivered randomly in intervals from 3 to 6 s. **b** Grand average of the late component of the sensory response of dMSNs in control (left) and 6-OHDA (right) lesioned mice to contralateral (dark blue traces) and ipsilateral (light blue traces) whisker stimulation. The light blue rectangle indicates the late component of the response to sensory stimulation. The gray bar indicates the moment the stimulation was delivered. **c** No differences in response amplitude, onset delay or area under the curve (AUC) of dMSNs responses to contralateral (C) and ipsilateral (I) stimulations in control or 6-OHDA lesioned mice. Note the diminution in response amplitude (dMSN control C = 3.85 ± 0.73 mV; dMSN 6-OHDA C = 2.06 ± 0.45 mV, two-sided Mann-unpaired t

test, * P = 0.01; dMSN control I = 4.01 ± 0.87 mV; dMSN 6-OHDA I = 1.64 ± 0.40 mV, two-sided unpaired t test, * P = 0.02) without affecting onset delay or AUC (n = 13) in the dMSNs of 6-OHDA lesioned mice compared to dMSNs in control mice (n = 13). **d** Grand average of the late component of iMSNs in control (left) and 6-OHDA (right) lesioned mice to contralateral (red) and ipsilateral (light red) whisker stimulation. The light red rectangle indicates the late component of the response to sensory stimulation. The gray bar indicates the moment the stimulation was delivered. **e** No differences in response amplitude, onset delay or AUC of iMSNs between contralateral (C) and ipsilateral (I) stimulations in control (n = 12) or 6-OHDA lesioned mice (n = 13). For **c**, **e** each gray line represents the data from a single cell and error bars indicate mean ± SEM. Complete summary of values is shown in Supplementary Table 2. Source data are provided as a Source Data file.

information and requests for resources and reagents should be directed to and will be fulfilled by the corresponding author. Source data are provided with this paper.

## Code availability
Analysis was done using standard built-in functions in Prism 8, Matlab and DeepLabCut.

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

## Acknowledgements

We thank Elin Dahlberg and Kristoffer Tenebro Berglund for technical assistance, members of the Silberberg lab for helpful discussions. We thank Ramon Reig, Abdel El Manira, and Sten Grillner for comments on earlier versions of the manuscript. We are thankful to Ole Kiehn for the Ai32 mice and to Gilberto Fisone for the D2-Cre mice. This work was supported by a Wallenberg Fellowship from the Knut & Alice Wallenberg Foundation (KAW 2017.0273), the Swedish Brain Foundation (Hjärnfonden, FO2021-0333), and the Swedish Medical Research Council (V.R.-M., 2019-0 1254) to G.S.. M.K. and R.d.l.T.M. were supported by grants from the Strategic Area in Neuroscience at KI (StratNeuro) and R.d.l.T.M. was supported by a Karolinska Institutet postdoctoral scholarship.

## Author contributions

R.d.l.T.M, M.K., and G.S. conceived and planned the experiments. M.K. performed the 6-OHDA lesions, R.d.l.T.M. performed the experiments and analyzed the data. R.d.l.T.M, M.K., and G.S. wrote the manuscript.

## Funding

## Competing interests

The authors declare no competing interests.
