## [Peer Review File · Nature Communications]

Ongoing movement controls sensory integration in the dorsolateral striatum.REVIEWER COMMENTS

Reviewer #1 (Remarks to the Author):

The paper by De La Torre Martinez and colleagues deals with sensorimotor interactions in the mouse dorsal striatum. Specifically, it aims to answer how whisking shapes sensory responses to tactile stimulation, in the striatum of control and Parkinsonian mice.

The method used is striatal whole-cell recordings in awake mice using the optopatcher for classifying direct and indirect pathway neurons. The authors show that the impact of movement on striatal sensory responses is, in all tested cases, attenuation. They further suggest one mechanism for the attenuation, based on the increase of the membrane conductance of neurons during movements. In addition to this main general question, the study compares sensory and motor representations in control and Parkinsonian (unilateral 6OHDA dopamine depletion model) mice, showing alterations in both sensory and motor responses.

The paper is interesting and timely, providing novel findings regarding sensory-motor interactions in the basal ganglia. The dataset is unique for the field, obtained using a challenging experimental method (deep whole-cell patch-clamp recordings in awake mice), for which the authors should be congratulated. The methods, experimental design as well as the presentation of the results and analysis are clear and excellent. However, there are a few shortcomings in the analysis, presentation, and necessary controls. The paper could be significantly improved by addressing these specific comments:

MAJOR:

1. The story can become much more interesting by presenting their results more in the context of the proposed function of basal ganglia in health and disease. In particular, they should examine the effect of sensory stimulation on initiation of movement. Previous studies demonstrated that mice start whisking just before they start locomoting. Although in their experimental conditions mice were unable to walk or run (perhaps their body was restrained but this is not fully explained in the methods), whisking provides an excellent indication for the intention of the animal to move. Hence, analysis of whisking behavior should be added, best as the starting point for the study. This clearly will attract more attention from the large community of system neuroscientists. I present more specific on this issue below (point 3 below).
2. The air-puff protocol is often used as an aversive stimulus when aimed to the face of the animal. Moreover, the whisker stimulation is not as precise as could be obtained using a piezo/galvo stimulation system. Lastly, could there be a contralateral effect of the air-puff that could explain the bilateral responses? Although this is very unlikely, as ipsilateral responses are as large in some cells as for the contralateral response, this should be addressed. Hence, the authors should address these questions specifically in the methods section and if possible provide video recordings of examples of the ipsi/contra/bilateral stimulations to show that air puff on one side did not cause apparent movements of whiskers on the other side of the mouse snout.

3. The authors claim that whisking in the lesioned mice is similar to control mice in terms of the percentage of time spent whisking compared to quiescent. However, one of the clear findings based on their results is that whisker stimulation often induces movement. How was this “triggering” of whisker movement affected in the parkinsonian animals? This is an important issue that is ignored here and could have a direct bearing on the opposite interaction: not only the impact of movement on sensory integration, but also the impact of sensory input on triggering movement (control and in parkinsonian). As explained in point (1) above, analysis of this matter can potentially increase the impact of this study. Please also refer to the recent paper from Carandini lab (‘Behavioral origin of sound-evoked activity in mouse visual cortex’ in BioRxiv).

4. The authors propose that the sensory attenuation during whisking is due to the increase in membrane conductance during whisking. While this is indeed shown, the authors do not explore additional mechanisms that could support the attenuation. One such possibility which is discussed but not tested is that the attenuation is inherited from the barrel cortex. It has been shown previously that whisking attenuates sensory responses in barrel cortex under certain conditions, as mentioned by the authors (Crochet & Petersen, 2006). While attenuation at the thalamic and brainstem level would be harder to show, the cortical recordings are essential, being the main input driving the striatal responses. The authors should test this possibility and analyze their cortical LFP recordings that were acquired simultaneously with the striatal recordings.

5. In figure 4, which is the main figure describing the sensory attenuation, the division between the Q and W conditions should be clearer. The different cases should be explicitly marked, better examples should be provided, or at least with different scaling, allowing for clear differentiation between the different scenarios.

6. Related to the point above. The authors claim that MSNs start depolarize before the onset of motor activity. The defined whisking epochs by finding the crossing point above 1.5 times of the signal STD. First, it is not clear if the STD was measured from the entire whisking signal or from quiescent periods. Second, it is well possible that the whisking events start before the starting time, identified by the threshold and thus it is not clear if membrane potential indeed starts to depolarize before the initiation of whisking. Additional analysis can clear this issue. For example, the authors can use cross-correlation analysis between Vm and W or use Granger causality analysis.

7. Related to (6) above: is there any correlation between the amount of depolarization and the amount of whisking? In other words, do cells tend to depolarize more when mice exhibit greater whisking?

The late component of the response is analyzed in figures 7 and supp-4 and is suggested to be caused by the motor activity (whisking) triggered by the air-puff. Firstly, the late component is not fully abolished when separating the QQ vs. QW cases (supp fig 4b), and secondly, in previous papers by the same lab, there is a similar late component also in anesthetized mice. The authors should address this issue in their discussion and comparisons with previous studies and perhaps present other possibilities underlying the late component. Do the authors have videos showing cases of air-puffs triggering and not triggering whisker movement?

8. Although not crucial for the conclusions of this study, I am puzzled from the fact that the authors didn't try to map the receptive fields of the cells by stimulating individual whiskers using actuators or galvo stimulators. Since such an approach in awake mice might be difficult due to triggered movements

following whisker stimulation, I will not be surprised to hear from the authors that they could not obtain clear results from such mapping.

MINOR:

- In figure 1b the image of the fluorescent D2 sagittal section is blurry and should be replaced.
- Figure 1d-e, the membrane voltage should be marked at least partly with a line, as in 1f-g
- When using the optopatcher, was there any recording of IPSPs in ChR2-negative cells? This may be expected due to MSN-MSN connectivity.
- The examples for ipsilateral whisker stimulus in fig supp 2a are identical to those shown in figure 4a for the contralateral stimuli. Even if this is just an illustration, the examples should be of actual experiments, in particular since the same examples are used in 4b and 4c to show Q and W contralateral responses in dMSNs. Please present another example.

Reviewer #2 (Remarks to the Author):

The manuscript by Torre-Martinez et al. presents the results from a study employing in vivo whole-cell recordings of identified medium spiny neurons (MSNs) in the dorsolateral striatum of awake head-restrained mice with or without a unilateral 6-OHDA lesion of dopaminergic neurons. The authors correlate the activity of direct and indirect pathway MSNs with whisking and the responses to whisker stimulation. There are several conclusions drawn by the authors from this set of experiments. First, the authors conclude that there was sublinear summation of whisker-related sensory and motor signals in MSNs. Second, loss of the striatal dopaminergic innervation attenuated the representation of whisking in selectively in direct-pathway MSNs. Third, the lesion of dopaminergic neurons attenuated the differences in the sensory representation of ipsilateral and contralateral whiskers in both types of MSN.

Although the data is quite beautiful and hard won (these experiments are capable of being performed by only a handful of labs in the world), it is essentially all phenomenological. While this is a reasonable point of departure, without a rigorous delineation of the mechanisms underlying the phenomena described, the conceptual impact of the study on the Parkinson's disease field is very limited.

Other major concerns:

- Every movement is accompanied by a sensory signal. Moreover, sensory signals can precede discernible movement. How do we know that the nominally movement-related depolarization of MSNs

is not sensory? Conversely, the late 'sensory' component could have a motor origin, just not one related to whisking (as stated by the authors at the end of the Discussion).

- The sublinearity of nominal sensory and motor input to MSNs is hardly surprising – as pointed out by the authors. What experiment could be done to assess why this is the case? Why is it important?
- The characterization of the 6-OHDA lesion is not adequate. The authors need to clearly state the behavioral criterion used for inclusion/exclusion of mice. Moreover, it is well known that the striatal adaptations following 6-OHDA lesioning do not stabilize for 3-4 wks (there is an old literature on this point but see Rentsch et al. 2019 for a revisitation). The reliance upon mice with 2 wk survival times complicates the interpretation of the data.
- It is surprising that 6-OHDA lesioning did not affect any aspect of whisking. Certainly, this would not have been the case had other aspects of movement been examined (e.g., forelimb use). Why? Although the total time spent whisking between DA-depleted and control mice was not different (Fig S1), individual whisking epochs might be different in duration and magnitude. If indeed whisking is unaffected by lesioning, it strains the attempt to link impaired recruitment of dMSNs to locomotion – which clearly is impaired by lesioning. It would seem that dMSNs have nothing to do with controlling whisking.
- The apparent attenuation of contralateral sensory input to MSNs following lesioning is interesting, but not explored. Moreover, the discussion of this point is astonishingly brief given the literature showing alterations in the functional connectomes of both types of MSN following dopamine depletion.

Minor concerns:

- Tests of normality are unreliable with small (<10) sample sizes. Non-parametric statistics should be used for display and hypothesis testing in this situation.
- All of the recordings are made using the whole cell mode, which has the advantage of allowing sub-threshold events to be monitored but also can create dialysis artifacts. One potential artifact stems from simply altering normal ionic gradients. In these experiments, 'resting' MSNs have membrane potentials roughly between -60 and -70 mV, quite far away from the nominal K⁺ equilibrium potential in these neurons. This is not particularly surprising given that the cells are undoubtedly being subjected to ongoing GABAergic and glutamatergic input. But it is important to try to ensure that these inputs are disrupted as little as possible. Looking at the recording internal, it has a very low internal Cl⁻ concentration (5 mM, unless Cl⁻ was added during the pH correction). This should push the Cl⁻ reversal potential much more negative than is physiological (see Bracci et al.).
- Given that recorded neurons were filled with biocytin, it's unfortunate that more detailed localization data and cellular anatomy are not presented. This might help with the interpretation of the results. Previous work with this model has shown that there are changes in MSN synaptic connectivity that should be reflected in function.
- The authors should look at the correlation between whisking parameters and V_m. It also would be helpful to add heatmaps of whisking activity to Fig 3 to show that whisking is similar while V_m

depolarization is reduced in 6-OHDA dMSNs comparing to control dMSNs, at least using an averaged whisking trace with scale bar.

- In contrast to what is stated in the Results and shown in Suppl Fig 4b, from Suppl Fig 4c table, neither the amplitude nor AUC from QW groups were larger than those from corresponding QQ groups for dMSNs and iMSNs. Where is the statistical analysis?

Reviewer #3 (Remarks to the Author):

In this study, de la Torre-Martinez and colleagues examine the impact of motor activity in sensory information processing within the dorsal striatum in control and parkinsonian mice. For this purpose, the authors used self-generated whisking as the motor spontaneous “stimulation” to evaluate their effect (as previously demonstrated in the cerebral cortex) in dorsolateral striatum (DLS) medium spiny neurons (MSNs) with the use of in vivo whole-cell recordings in awake and head-restrained mice. In addition, the authors were able to identify MSNs belonging to the indirect (i-MSNs) or the direct (d-MSNs) trans-striatal pathway by using D1-cre and D2-cre mice expressing Ai32 channelrhodopsine; d-MSNs and i-MSNs were then opto-identified with the optogenetic evoked-response to blue light pulses via the opto-patcher device. At this point, one should acknowledge that these experiments represent a genuine technical tour-de-force and that the electrophysiological recordings are of high quality.

The main findings of this study are that whisker deflection evoked-responses in MSNs were attenuated during self-generated whisking (with most of the MSNs responding to both spontaneous motor activity and evoked sensory stimulation), and that in parkinsonian mice (6-OHDA/MFB model) both motor and sensory responses were affected predominantly in the d-MSNs.

Overall, this constitutes a very impressive amount of work, providing (elegant) in-depth analysis of the intersection between motor and sensory information processing in the DLS in control and parkinsonian mouse model. The resulting data are very interesting and novel. This study brings a novel (and highly interesting) piece to the interaction of sensory and motor information at the level of the striatum despite a lack of mechanistic explanation.

This study is undoubtedly of significance to the field and related fields.

Major comment:

- In Fig. 3 could the effect between d- and i-MSNs could result from the relative imbalance in the number of recorded d-MSNs and i-MSNs (n= 8 and 17, respectively) in DA-depleted mice?

Ideally, the authors should increase the number of d-MSNs recorded in DA-depleted mice.

- Fig. 5. The mechanism(s) accounting for the attenuation of sensory responses by ongoing self-generated motor activity (whisking) could be better documented here. The authors very nicely demonstrate that a reduction of input resistance of (one?) i-MSNs is occurring during spontaneous whisking is responsible for this attenuation. However, it is not clear in how many MSNs (just one i-MSNs?) this was observed. Is this conclusion stand also in d-MSNs? I understand that the experiments are challenging but this is a key point of the study, and one could expect few i-MSNs and d-MSNs recorded to asses proper Ri characterization during quiescent and whisking epochs (in control condition at least).

In the same line, the authors should discuss also other mechanisms (not intrinsic membrane properties) that could be at play such as GABAergic interneuron activity and others.

Minor comments:

- Line 94. Please give the % of DA depletion in the 6-OHDA mice.

- Lines 98-103: please also report and comment the (absence of?) difference between d-MSNs and i-MSNs in terms of RMP and Ri.

- In Fig. 2a, it is not clear what the grey rectangle in the right hemisphere stands for.

- The Supplementary Fig.2 brings important information. The authors could (but up to them) integrate it in the Fig4 or put it as a main figure by itself.

- Fig.7 legend: the abbreviation AUC is defined twice (lines 559 and 561).

- Supplementary Fig. 1: could you indicate in the figure the stats between control and DA-depleted conditions for Q and W epochs?

- Line 171.The Supplementary Fig.3 should be better explained.

- Use the self-defined abbreviations throughout the manuscript: DLS (for example: title of Fig. 1 or XXXX), DA.

REVIEWER COMMENTS

Reviewer #1 (Remarks to the Author):

The paper by De La Torre Martinez and colleagues deals with sensorimotor interactions in the mouse dorsal striatum. Specifically, it aims to answer how whisking shapes sensory responses to tactile stimulation, in the striatum of control and Parkinsonian mice.

The method used is striatal whole-cell recordings in awake mice using the optopatcher for classifying direct and indirect pathway neurons. The authors show that the impact of movement on striatal sensory responses is, in all tested cases, attenuation. They further suggest one mechanism for the attenuation, based on the increase of the membrane conductance of neurons during movements. In addition to this main general question, the study compares sensory and motor representations in control and Parkinsonian (unilateral 6OHDA dopamine depletion model) mice, showing alterations in both sensory and motor responses.

The paper is interesting and timely, providing novel findings regarding sensory-motor interactions in the basal ganglia. The dataset is unique for the field, obtained using a challenging experimental method (deep whole-cell patch-clamp recordings in awake mice), for which the authors should be congratulated. The methods, experimental design as well as the presentation of the results and analysis are clear and excellent. However, there are a few shortcomings in the analysis, presentation, and necessary controls. The paper could be significantly improved by addressing these specific comments:

We thank the reviewer for these positive comments and interest in our work. We have changed the study significantly by adding experiments, analyses, and modifying figures and text in the revised version. Below is a point-by-point reply to all the reviewer's comments, which we hope will be satisfactory.

MAJOR:

1. The story can become much more interesting by presenting their results more in the context of the proposed function of basal ganglia in health and disease. In particular, they should examine the effect of sensory stimulation on initiation of movement. Previous studies demonstrated that mice start whisking just before they start locomoting. Although in their experimental conditions mice were unable to walk or run (perhaps their body was restrained but this is not fully explained in the methods), whisking provides an excellent indication for the intention of the animal to move. Hence, analysis of whisking behavior should be added, best as the starting point for the study. This clearly will attract more attention from the large community of system neuroscientists. I present more specific on this issue below (point 3 below).

We thank the reviewer for this comment, and we have now extended the analysis of the whisking patterns in control and 6-OHDA mice. The new analyses include quantification of the durations and frequencies of spontaneous whisking bouts in control and DA depleted mice. In addition, as suggested by the reviewer, we analyzed the whisking evoked by whisker stimulation, thus providing insight about the sensory-motor transformation in control and dopamine-depleted

mice. The results are now presented in Supplementary Fig. 1, and described in the text in lines 94-105. Sensory stimulation triggered whisking in most trials and under all conditions (ipsi/contra and control/6-OHDA). There was a trend indicating a slightly lower probability for evoking movement in lesioned animals, however, these differences were not significant (Supplementary Fig. 1). Regarding the relationship between whisking and locomotion, this was not studied here since in our experimental conditions the mouse is restrained in a tube and not performing any locomotive movement. The only motor aspect we studied and quantified was the whisking. This is now described in further detail in the methods section (lines 94-105, 232-235, 391-435, in the revised MS)

2. The air-puff protocol is often used as an aversive stimulus when aimed to the face of the animal. Moreover, the whisker stimulation is not as precise as could be obtained using a piezo/galvo stimulation system. Lastly, could there be a contralateral effect of the air-puff that could explain the bilateral responses? Although this is very unlikely, as ipsilateral responses are as large in some cells as for the contralateral response, this should be addressed. Hence, the authors should address these questions specifically in the methods section and if possible provide video recordings of examples of the ipsi/contra/bilateral stimulations to show that air puff on one side did not cause apparent movements of whiskers on the other side of the mouse snout.

To address these comments and suggestions, we have now performed control experiments in which we filmed mice during whisker stimulation and analyzed the impact of air-puffs on the contralateral side and eyes using DeepLabCut (Methods section, lines 417-435). The reviewer raised concerns that the air-puff would 1) be an aversive stimulus and 2) affect the contralateral whiskers. In our experimental setup we took extra care to avoid both cases. The air-puffs were directed to the distal parts of the whiskers and not to the snout or face of the mouse, and moreover, were very brief. In order to demonstrate the lack of aversion we now present a video (Supplementary video 2) showing whisker stimulation in an awake mouse while monitoring the ipsilateral eye, showing that there was no triggering of an eye-blink, which indicates an aversive stimulation. We also present a video showing ipsi- contra- and bilateral whisker stimulation in an anesthetized mouse, where whiskers are tracked using DeepLabCut, demonstrating the lack of contralateral interactions of the stimuli (Supplementary video 1). As can be seen in the video clip, unilateral stimulation evoked a brief deflection, detectable only in the ipsilateral whiskers but not in the contralateral ones. In awake mice, such unilateral air-puffs often triggered centralized movement in the form of bilateral whisking bouts (Supplementary video 2).

3. The authors claim that whisking in the lesioned mice is similar to control mice in terms of the percentage of time spent whisking compared to quiescent. However, one of the clear findings based on their results is that whisker stimulation often induces movement. How was this “triggering” of whisker movement affected in the parkinsonian animals? This is an important issue that is ignored here and could have a direct bearing on the opposite interaction: not only the impact of movement on sensory integration, but also the impact of sensory input on triggering

movement (control and in parkinsonian). As explained in point (1) above, analysis of this matter can potentially increase the impact of this study. Please also refer to the recent paper from Carandini lab ('Behavioral origin of sound-evoked activity in mouse visual cortex' in BioRxiv). To address this point, we re-analyzed our data to quantify the degree of sensory-evoked whisking under the different experimental conditions. The summary is presented here and in the revised MS (Supplementary Fig. 1), showing the high degree (>90%) of evoked whisking by unilateral whisker stimulation in all cases. We found that in 6-OHDA lesioned mice there was a slight yet non-significant decrease in the probability for evoking whisking bouts, suggesting a minor role of striatal DA in mediating sensory-evoked whisking. With regard to the Carandini paper mentioned by the reviewer, indeed, sensory input from different modalities could induce whisking in mice. In our study we only tested whisker stimulation evoked whisking. We have previously shown that neurons in the dorsal striatum responded to visual stimuli as well (Reig & Silberberg, 2014), yet this was only done in anesthetized mice and not expanded to awake mice in this study. We added a section in the discussion pertaining to this point (lines 232-235).

4. The authors propose that the sensory attenuation during whisking is due to the increase in membrane conductance during whisking. While this is indeed shown, the authors do not explore additional mechanisms that could support the attenuation. One such possibility which is discussed but not tested is that the attenuation is inherited from the barrel cortex. It has been shown previously that whisking attenuates sensory responses in barrel cortex under certain conditions, as mentioned by the authors (Crochet & Petersen, 2006). While attenuation at the thalamic and brainstem level would be harder to show, the cortical recordings are essential, being the main input driving the striatal responses. The authors should test this possibility and analyze their cortical LFP recordings that were acquired simultaneously with the striatal recordings.

We followed this suggestion and analyzed the cortical LFP recordings in S1, acquired simultaneously with the striatal whole-cell recordings. Indeed, we observed an attenuation in the amplitude of the cortical LFP when stimuli were delivered during whisking. This result is in agreement with the findings in the paper by Crochet & Petersen (2006) and shows that the cortical input mediating the striatal response is attenuated and is further diminished by the conductance changes in striatal MSNs. Both these synergistic mechanisms are now described in the revised MS (lines 153-163 and Fig. 5).

5. In figure 4, which is the main figure describing the sensory attenuation, the division between the Q and W conditions should be clearer. The different cases should be explicitly marked, better examples should be provided, or at least with different scaling, allowing for clear differentiation between the different scenarios.

We have modified the figure (Fig. 5 in the revised MS) according to the comments, to emphasize the different stimulus scenarios.

6. Related to the point above. The authors claim that MSNs start depolarize before the onset of motor activity. The defined whisking epochs by finding the crossing point above 1.5 times of the signal STD. First, it is not clear if the STD was measured from the entire whisking signal or from quiescent periods. Second, it is well possible that the whisking events start before the starting time, identified by the threshold and thus it is not clear if membrane potential indeed starts to depolarize before the initiation of whisking. Additional analysis can clear this issue. For example, the authors can use cross-correlation analysis between Vm and W or use Granger causality analysis.

Regarding STD: we have now expanded the description in the methods section to clarify the threshold determination for whisking. The deviation by 1.5 STD was from a baseline level calculated from 2 s of quiescence whisker signal (Methods section, lines 391-405, 436-444). Secondly, we added an analysis of the relationship between whisking and MSN membrane depolarization by performing a cross-correlation between the membrane voltage and whisker sensor traces around the transition times, now presented in Supplementary Fig. 2-3 and text (lines 108-127). There is indeed a positive correlation between the two signals, and moreover, the peak of the cross-correlogram has a lag of 32 ms for dMSNs and 39 ms for iMSNs, similar to the delays shown in the examples in Fig. 3, further showing that the membrane depolarization precedes the actual whisker movement by tens of milliseconds.

7. Related to (6) above: is there any correlation between the amount of depolarization and the amount of whisking? In other words, do cells tend to depolarize more when mice exhibit greater whisking? The late component of the response is analyzed in figures 7 and supp-4 and is suggested to be caused by the motor activity (whisking) triggered by the air-puff. Firstly, the late component is not fully abolished when separating the QQ vs. QW cases (supp fig 4b), and secondly, in previous papers by the same lab, there is a similar late component also in anesthetized mice. The authors should address this issue in their discussion and comparisons with previous studies and perhaps present other possibilities underlying the late component. Do the authors have videos showing cases of air-puffs triggering and not triggering whisker movement?

We used a cross-correlation analysis to show the relationship between MSN depolarization and whisking under different conditions. This analysis is now described in the results section (lines 108-127) and presented in new supplementary figures, showing that the peak correlation, reflecting a strong relationship between membrane voltage and movement, was reduced specifically in dMSNs in 6-OHDA lesioned mice (Supplementary Fig. 2c). The correlation between the amount of depolarization and whisking intensity in individual whisking epochs is presented in supplementary Fig. 3, showing positive correlation between depolarization and whisking vigor, measured as the area under the curve of the rectified whisker trace.

Regarding the late component, the reviewer correctly points out that the stimulus-evoked whisking cannot fully explain the late component. Indeed, the late component is not fully abolished in the “QQ” cases but only reduced, and, moreover, it does appear also in anesthetized

mice, as we have shown earlier (Reig & Silberberg, 2014). This raises the possibility that part of the late component is generated by consequent movement of the whisker and/or recurrent neuronal activity triggered by the stimulus. We have now further clarified this point in the revised manuscript (lines 287-297).

8. Although not crucial for the conclusions of this study, I am puzzled from the fact that the authors didn't try to map the receptive fields of the cells by stimulating individual whiskers using actuators or galvo stimulators. Since such an approach in awake mice might be difficult due to triggered movements following whisker stimulation, I will not be surprised to hear from the authors that they could not obtain clear results from such mapping.

We agree with the reviewer that the mapping of the receptive fields of MSNs in terms of stimulation of individual whiskers is indeed a very interesting question. However, as also pointed out by the reviewer, our whisker stimulation method using air-puffs towards several whiskers during quiescence and whisking may not be the optimal method for answering this specific question. In order to stimulate individual whiskers using piezo or galvo devices we would have to interfere with spontaneous and evoked whisking, both of which are central in the current study. Utilization of unilateral and bilateral air-puffs enabled us to deliver the same stimuli during quiescence and spontaneous whisking. The trade-off in this approach is the specificity in terms of the individual whisker stimulation.

MINOR:

- In figure 1b the image of the fluorescent D2 sagittal section is blurry and should be replaced.

This issue has been corrected.

- Figure 1d-e, the membrane voltage should be marked at least partly with a line, as in 1f-g

This issue has been corrected.

- When using the optopatcher, was there any recording of IPSPs in ChR2-negative cells? This may be expected due to MSN-MSN connectivity.

The optopatcher was used only for identification of MSNs, allowing for a quick and reliable online way to determine the expression of ChR2 in recorded cells. The photostimulation was not especially strong and it is not clear whether and how many neighboring cells would be sufficiently excited to fire action potentials. We therefore did not search for MSN-MSN connectivity in our experiments, as these methods are clearly not optimal for answering this question. Another factor is the reversal potential for inhibition, which is very close to the resting membrane potential of MSNs in our experimental conditions (see more about this in responses to reviewer #2 and line 354 in the Methods for intracellular solution composition). In order to study optopatcher-evoked synaptic responses, we would have needed a large number of repetitions as well as holding cells at depolarized potentials, that would expose such interaction. As this was not the purpose of the experiments and recordings are relatively short we opted to perform a quick classification of MSNs and proceed as fast as possible to the sensory protocols. Regarding this point, also see (Ketzev et al, 2017), where we could show a few cases in which

synaptic potentials were evoked using the optopatcher *in vivo* (Fig. 1) and observe MSN-MSN connectivity *ex vivo* when providing wide-field photostimulation.

- The examples for ipsilateral whisker stimulus in fig supp 2a are identical to those shown in figure 4a for the contralateral stimuli. Even if this is just an illustration, the examples should be of actual experiments, in particular since the same examples are used in 4b and 4c to show Q and W contralateral responses in dMSNs. Please present another example.

This issue has been corrected.

Reviewer #2 (Remarks to the Author):

The manuscript by Torre-Martinez et al. presents the results from a study employing *in vivo* whole-cell recordings of identified medium spiny neurons (MSNs) in the dorsolateral striatum of awake head-restrained mice with or without a unilateral 6-OHDA lesion of dopaminergic neurons. The authors correlate the activity of direct and indirect pathway MSNs with whisking and the responses to whisker stimulation. There are several conclusions drawn by the authors from this set of experiments. First, the authors conclude that there was sublinear summation of whisker-related sensory and motor signals in MSNs. Second, loss of the striatal dopaminergic innervation attenuated the representation of whisking in selectively in direct-pathway MSNs. Third, the lesion of dopaminergic neurons attenuated the differences in the sensory representation of ipsilateral and contralateral whiskers in both types of MSN.

Although the data is quite beautiful and hard won (these experiments are capable of being performed by only a handful of labs in the world), it is essentially all phenomenological. While this is a reasonable point of departure, without a rigorous delineation of the mechanisms underlying the phenomena described, the conceptual impact of the study on the Parkinson's disease field is very limited.

We thank the reviewer for the thorough review and appreciation of our work. To address the reviewer's comments, in particular regarding underlying mechanisms, we have revised the manuscript by adding experiments, analyses, and modifying figures and text accordingly. Specifically, to address the last comment, we now highlight three different mechanisms that are directly linked to the attenuation of sensory responses during movement. Below is a point-by-point reply to all the reviewer's comments, which we hope will be satisfactory.

Other major concerns:

- Every movement is accompanied by a sensory signal. Moreover, sensory signals can precede discernible movement. How do we know that the nominally movement-related depolarization of MSNs is not sensory? Conversely, the late 'sensory' component could have a motor origin, just not one related to whisking (as stated by the authors at the end of the Discussion).

This is a valid point, and indeed, we cannot trace the origin of the pre-whisking depolarization that we see in MSNs at the onset of spontaneous whisking bouts. The situation is different when we deliver the tactile stimulus, and a whisking bout follows. In the cases of spontaneous

whisking depicted in Fig.3, we do not know what the internal or external triggers for the whisking bouts are, however, we do know that they were not triggered by the experimentally controlled whisker deflection. This is the only distinction made in our analysis. The depolarization reflects the whisking onset, and even precedes it by tens of milliseconds, as seen by two different analyses, now presented in Fig. 3 and Supplementary Fig. 2. We have now extended the analysis of the relationship between the MSN membrane voltage and the whisker activity during whisking onset using cross-correlation. This indeed shows a positive correlation, with peak time lag of 32 ms for dMSNs and 39 ms for iMSNs, of the membrane depolarization before movement onset. The cross-correlation analysis is presented in Supplementary Fig. 2,3 in the revised manuscript and described in the Methods (lines 436-444).

Regarding the late component of responses to whisker stimulation, indeed, as the reviewer points out, they could have a “motor component”. This is the reason we performed the “QQ” and “QW” analysis depicted in Supplementary Fig. 6. We show that part of the late component can be attributed to the whisking movement triggered by the stimulation. However, it is important to note that whisking alone does not fully account for the late depolarization. There might be other movements (trunk, paws, neck, face) that we do not monitor and, moreover, late response components to whisker deflections were also seen in anesthetized mice (Reig & Silberberg, 2014), suggesting a non-motor component as well. These points are now elaborated in the results and discussion sections (lines 199-205 and 287-297).

- The sublinearity of nominal sensory and motor input to MSNs is hardly surprising – as pointed out by the authors. What experiment could be done to assess why this is the case? Why is it important?

The regulation of sensory responses by movement is the central question in this study, and we thank the reviewer for pointing out that this was not sufficiently clear. Our results show that sensory synaptic responses in MSNs are attenuated when the mouse is engaged in whisking compared to sensory responses during quiescence. Similar experiments were previously done in cortical regions, showing attenuation in auditory and tactile responses, but enhancement of visual responses in V1 (Polack, Friedman, Golshani, *Nature Neuroscience*, 2013). In striatum, such experiments and analysis have not been carried out previously and we obtained and present a clear answer for this question.

Regarding the underlying mechanisms, we have revised this part significantly and added data from the simultaneously recorded cortical LFP. We now present two synergistic mechanisms underlying the observed response attenuation. One mechanism is “striatal”, in which the membrane conductance of MSNs increases during whisking, resulting in smaller voltage responses to the same input. Following the comments from reviewers #1 and #2 we analyzed the cortical LFP responses and show that also at the cortical level there is a whisking related attenuation, suggesting that the corticostriatal input is attenuated also at the presynaptic cortical level. Both mechanisms are now described in the results (lines 153-163) and in Fig. 5.

- The characterization of the 6-OHDA lesion is not adequate. The authors need to clearly state the behavioral criterion used for inclusion/exclusion of mice. Moreover, it is well known that the striatal adaptations following 6-OHDA lesioning do not stabilize for 3-4 wks (there is an old literature on this point but see Rentsch et al. 2019 for a revisitation). The reliance upon mice with 2 wk survival times complicates the interpretation of the data.

We thank the reviewer for this comment, and we now clarify the 6-OHDA protocol, timeline, and behavioral inclusion criteria for lesion assessment (see Methods lines 314-338). In summary, the experiments were done within 3-5 weeks following the 6-OHDA injections, and the effect of the lesion was assessed by analysis of rotational behavior of 6-OHDA injected mice. Indeed, the Rentsch paper describes the changes following the 6-OHDA lesion, showing that already after one week following injection there is an almost complete degeneration of striatal DA axons, which is the relevant aspect for our study (Figure 2 therein), and a longer time needed for somatic degeneration in the midbrain. As mentioned above, our recordings were performed within 3-5 weeks after the lesion, when the striatum is virtually devoid of DA axons, as seen in Fig. 2 in the revised manuscript (and in Rentsch 2019, Fig.2). The methods section was modified to clarify these aspects of the lesion.

- It is surprising that 6-OHDA lesioning did not affect any aspect of whisking. Certainly, this would not have been the case had other aspects of movement been examined (e.g., forelimb use). Why? Although the total time spent whisking between DA-depleted and control mice was not different (Fig S1), individual whisking epochs might be different in duration and magnitude. If indeed whisking is unaffected by lesioning, it strains the attempt to link impaired recruitment of dMSNs to locomotion – which clearly is impaired by lesioning. It would seem that dMSNs have nothing to do with controlling whisking.

The observation that the 6-OHDA lesion had no effect on the overall duration of whisking vs quiescent is indeed surprising, but of course does not mean that the lesion would not affect other motor programs such as trunk and paw movement, locomotion etc. In this study we focused on the sensorimotor aspects of the whisker system representation in striatum.

Following these comments, raised also by reviewer #1 we have further analyzed the whisking properties such as the frequency and duration of individual whisking bouts, now presented in Supplementary Fig. 1 and lines 94-99 in the revised manuscript. In this study as well as in previous work, the 6-OHDA lesion did not eliminate whisking, did not eliminate sensory responses to whisker stimulation, and had very small impact on the activity of individual MSNs in awake and anesthetized mice. The effects of DA depletion in our experimental conditions are subtle, despite the broad and almost complete dopamine lesion.

- The apparent attenuation of contralateral sensory input to MSNs following lesioning is interesting, but not explored. Moreover, the discussion of this point is astonishingly brief given the literature showing alterations in the functional connectomes of both types of MSN following dopamine depletion.

We thank the reviewer for this comment, and we now expanded the discussion regarding this point. The reduction in “bilateral encoding”, as we refer to the differences between responses to ipsi- and contralateral whisker deflections, is indeed an interesting phenomenon and we have described it both in anesthetized mice (Ketzer et al, 2017) and in this study with awake mice. Although we do not show a mechanism underlying this change, we now raise a few hypotheses that could be investigated in future studies. One possibility is that the DA lesion induces differential changes in cortical circuits, such as the PT and IT corticostriatal and cortico-callosal (also mediated by IT pyramidal cells) pathways. Such changes could arise from compensatory processes following the lesion, strengthening specific synapses while attenuating others. Another possibility is that changes in the intrinsic and synaptic properties of striatal neurons affect the bilateral inputs. Specifically, cholinergic interneurons (ChINs) are innervated by the PT and not IT corticostriatal pathway (Johansson & Silberberg, 2020, Morgenstern et al, 2022), suggesting that changes in ChIN properties would mainly affect PT mediated input. Moreover, ChINs affect striatal afferent excitatory synaptic transmission (Pakhotin & Bracci, 2007) but it is not known whether this modulation is uniform or whether it targets specific pathways and not others. These are interesting questions but would require further studies beyond the scope of the current paper. As suggested by the reviewer, we have now expanded the discussion about this issue (lines 275-286).

Minor concerns:

- Tests of normality are unreliable with small (<10) sample sizes. Non-parametric statistics should be used for display and hypothesis testing in this situation.

We have addressed this comment in some of the data. In Fig. 3 and Fig. 4, in the figure legends, we now include the comparison that is done with parametric tests and with nonparametric tests. We also mentioned which one we use. The other comparisons are not treated as nonparametric since they are larger than 10 and do follow a normal distribution.

- All of the recordings are made using the whole cell mode, which has the advantage of allowing sub-threshold events to be monitored but also can create dialysis artifacts. One potential artifact stems from simply altering normal ionic gradients. In these experiments, ‘resting’ MSNs have membrane potentials roughly between -60 and -70 mV, quite far away from the nominal K⁺ equilibrium potential in these neurons. This is not particularly surprising given that the cells are undoubtedly being subjected to ongoing GABAergic and glutamatergic input. But it is important to try to ensure that these inputs are disrupted as little as possible. Looking at the recording internal, it has a very low internal Cl⁻ concentration (5 mM, unless Cl⁻ was added during the pH correction). This should push the Cl⁻ reversal potential much more negative than is physiological (see Bracci et al.).

The reviewer raises an important point, which is the impact of the pipette solution in whole cell recordings. Specifically, the Chloride concentration in the pipette will have a direct impact on the reversal potential of GABA_A mediated inputs. We used an internal solution containing 5 mM

Cl⁻ which results in a reversal potential of ~ -70 mV, and a liquid junction potential of ~ 11 mV was not corrected (see methods section, lines 352-367). The solution was tested both *in vivo* and in slices and enabled long and stable recordings. In comparison, *in vivo* striatal recordings performed by a different group (Petersen lab) using 4 mM Chloride reported resting potentials between -74 and -71 mV, suggesting the GABA_A synaptic transmission indeed affects the resting membrane potential in striatal *in vivo* awake recordings. Previous work done using sharp recordings (1 or 2 Molar potassium based solutions) reported more hyperpolarized resting potentials (Mahon, 2001, Schulz and Reynolds studies), which could be due to the recording method with sharp electrodes and also the arousal conditions. Indeed, under ketamine or urethane anesthesia, the membrane potential of MSNs exhibits transitions between down- and up-states. The membrane potential in awake recordings typically lies between the down- and up-state potentials, with the down-state potentials being more hyperpolarized and closer to the Potassium reversal potential. Therefore, given the pipette Chloride concentration and the awake state of the mice in our recordings, a resting potential between -60 and -70 mV is not surprising and rather close to physiological conditions. One possibility is to perform perforated patch recordings *in vivo*, but this would make these already difficult recordings much harder and shorter, thus reducing the yield significantly.

- Given that recorded neurons were filled with biocytin, it's unfortunate that more detailed localization data and cellular anatomy are not presented. This might help with the interpretation of the results. Previous work with this model has shown that there are changes in MSN synaptic connectivity that should be reflected in function.

We completely agree that it would be good to have unambiguous localization of recorded neurons, however, this would prove to be impossible since we recorded from several neurons in the same striatal hemisphere. This was our main motivation for using the optopatcher, enabling us to record from more than one cell per animal. The downside of this approach is that we only have the rough location of recordings in the dorsolateral striatum but not the exact locations and morphological reconstructions of recorded neurons. The coordinates in dorsolateral striatum for all recordings are specified in the methods (lines 346-352) "(0 mm A-P; 3 mm M-L) from 2 to 2.5 mm below the pia". The locations of selected recorded and recovered cells are presented in the figure below:

- The authors should look at the correlation between whisking parameters and V_m . It also would be helpful to add heatmaps of whisking activity to Fig 3 to show that whisking is similar while V_m depolarization is reduced in 6-OHDA dMSNs comparing to control dMSNs, at least using an averaged whisking trace with scale bar.

We thank the reviewer for this comment, which is shared with reviewer #1. We have now added a cross-correlation analysis between the membrane voltage and whisking activity, showing positive correlation between the two, and moreover, showing a peak lag of 32 ms for dMSNs and 39 ms for iMSNs, indicating the preceding membrane depolarization before whisking onset (Supplementary Fig. 2). We also show the relationship between the depolarization and whisking intensity for individual whisking bouts, showing a positive correlation (Supplementary Fig. 3). Lastly, we have modified Fig. 3 in the revised MS according to the reviewer's suggestion, showing the corresponding sweeps of whisking activity as a heatmap.

- In contrast to what is stated in the Results and shown in Suppl Fig 4b, from Suppl Fig 4c table, neither the amplitude nor AUC from QW groups were larger than those from corresponding QQ groups for dMSNs and iMSNs. Where is the statistical analysis?

Corrected in revised manuscript.

Reviewer #3 (Remarks to the Author):

In this study, de la Torre-Martinez and colleagues examine the impact of motor activity in sensory information processing within the dorsal striatum in control and parkinsonian mice. For this purpose, the authors used self-generated whisking as the motor spontaneous “stimulation” to evaluate their effect (as previously demonstrated in the cerebral cortex) in dorsolateral striatum (DLS) medium spiny neurons (MSNs) with the use of in vivo whole-cell recordings in awake and head-restrained mice. In addition, the authors were able to identify MSNs belonging to the indirect (i-MSNs) or the direct (d-MSNs) trans-striatal pathway by using D1-cre and D2-cre mice expressing Ai32 channelrhodopsine; d-MSNs and i-MSNs were then opto-identified with the optogenetic evoked-response to blue light pulses via the opto-patcher device. At this

point, one should acknowledge that these experiments represent a genuine technical tour-de-force and that the electrophysiological recordings are of high quality.

The main findings of this study are that whisker deflection evoked-responses in MSNs were attenuated during self-generated whisking (with most of the MSNs responding to both spontaneous motor activity and evoked sensory stimulation), and that in parkinsonian mice (6-OHDA/MFB model) both motor and sensory responses were affected predominantly in the d-MSNs. Overall, this constitutes a very impressive amount of work, providing (elegant) in-depth analysis of the intersection between motor and sensory information processing in the DLS in control and parkinsonian mouse model. The resulting data are very interesting and novel. This study brings a novel (and highly interesting) piece to the interaction of sensory and motor information at the level of the striatum despite a lack of mechanistic explanation. This study is undoubtedly of significance to the field and related fields.

We thank the reviewer for the comments and positive interest in our study. We address all the comments below in the point-by-point reply. Specifically, we have addressed the mechanistic aspects to further explain our findings.

Major comment:

- In Fig. 3 could the effect between d- and i-MSNs could result from the relative imbalance in the number of recorded d-MSNs and i-MSNs (n= 8 and 17, respectively) in DA-depleted mice? Ideally, the authors should increase the number of d-MSNs recorded in DA-depleted mice.

The recordings are performed blindly, and we cannot preselect or target recordings to dMSNs. In our dataset for this work, we happened to record from more iMSNs than dMSNs, however, the numbers for all groups are sufficient to detect significant differences between them. In order to accommodate the reviewer's comment, we performed more experiments and slightly increased the number of dMSNs by performing additional recordings in 6-OHDA lesioned mice.

- Fig. 5. The mechanism(s) accounting for the attenuation of sensory responses by ongoing self-generated motor activity (whisking) could be better documented here. The authors very nicely demonstrate that a reduction of input resistance of (one?) i-MSNs is occurring during spontaneous whisking is responsible for this attenuation. However, it is not clear in how many MSNs (just one i-MSNs?) this was observed. Is this conclusion stand also in d-MSNs? I understand that the experiments are challenging but this is a key point of the study, and one could expect few i-MSNs and d-MSNs recorded to assess proper Ri characterization during quiescent and whisking epochs (in control condition at least).

In the same line, the authors should discuss also other mechanisms (not intrinsic membrane properties) that could be at play such as GABAergic interneuron activity and others.

We thank the reviewer for this comment, which is also in line with similar comments from the other reviewers. The conductance experiment was performed in several neurons, both dMSNs and iMSNs. Only one was shown as an example. We hope that the revised Fig. 5c clarifies this issue.

To investigate additional mechanisms, we have now analyzed the cortical LFP recordings during quiescence and whisking, and we show that in addition to the postsynaptic striatal mechanism (conductance change during whisking) there is also a synergistic attenuation of cortical responses, suggesting that the corticostriatal input is also reduced on the presynaptic side. Both mechanisms support attenuation, and while the cortical attenuation was shown previously (Poulet & Petersen, 2006), we show the additional attenuation at the striatal level. Both mechanisms are now presented in the main Fig. 5 in the revised manuscript. Regarding MSN identity, the example given was of an iMSN but the data was collected from both dMSNs and iMSNs, which is now clarified in the text (lines 153-163) and the revised Fig. 5. We also address the possibility that the striatal circuitry including the different interneurons, could affect the MSN responses by changing their activity during quiescence and whisking (lines 247-260). Unfortunately, this is currently only speculative as we did not obtain recordings from interneurons in this study.

Minor comments:

Please give the % of DA depletion in the 6-OHDA mice.

TH staining showed over 80% reduction in fluorescence compared to unlesioned hemispheres (lines 389-390 in Methods section).

- Lines 98-103: please also report and comment the (absence of?) difference between d-MSNs and i-MSNs in terms of RMP and Ri.

This is modified in revised MS (lines 88-106 and Fig. 2).

- In Fig. 2a, it is not clear what the grey rectangle in the right hemisphere stands for.

The rectangle represents the head fixation plate. It is now described in the figure legend.

- The Supplementary Fig.2 brings important information. The authors could (but up to them) integrate it in the Fig4 or put it as a main figure by itself.

The supplementary figure shows that for ipsilateral whisker stimulation the results are following the same trend, with motor related attenuation. Adding it to Fig. 4 would make the figure too packed, without adding much information. Another reason for adding it as a supplementary figure is that the primary stimulation was the contralateral whisker, and we have more data points for this protocol. The ipsilateral stimulation presented in Supplementary Fig. 4, although showing a similar result, is based on less data points.

- Fig.7 legend: the abbreviation AUC is defined twice (lines 559 and 561).

This issue has been corrected.

- Supplementary Fig. 1: could you indicate in the figure the stats between control and DA-depleted conditions for Q and W epochs?

We have modified this figure (Supplementary Fig. 1) and added several new analyses comparing the whisking patterns in lesioned mice. Whereas there was no difference in the total whisking

duration between control and 6-OHDA mice, there were differences in the frequency and duration of whisking bouts.

- Line 171. The Supplementary Fig.3 should be better explained.

This issue has been corrected.

- Use the self-defined abbreviations throughout the manuscript: DLS (for example: title of Fig. 1 or XXXX), DA.

This issue has been corrected.

REVIEWER COMMENTS

Reviewer #1 (Remarks to the Author):

The authors nicely addressed point by point most of my concerns. I now strongly support the paper for publication. However, I would like to see the response of the authors to my minor comments as shown below.

1) I was 'disappointed' to see in Supplementary Figure 1 that whisking parameters were barely different for 6-OHDA mice compared to control mice as I expected to see some effect on behavior - but these are the results. In particular, the lack of change in initiation of whisking when comparing the 6-OHDA mice to control mice.

2) Yet, did the author find any difference in the duration of whisking following tactile stimulation across these two groups?

3) Therefore, I suggest to slightly change the last sentence of the abstract ("Our results show that motor activity affects sensory responses in basal ganglia circuits and that both processes are dopamine and cell type-dependent.") - first change 'motor' to 'whisking' or 'whisking related motor' and also since the whisking parameters were not dopamine dependent it is hard to understand the "both processes are dopamine ..." (the cell type is fine). The authors show that the membrane potential became more depolarized just before whisking. However, it is hard to know from this study if the MSN cells are involved in generation of whisking or just represent motor commands and proprioceptive inputs. Perhaps this should be more emphasized in the discussion.

4) The cross-correlation analysis strongly supports the speculation that Vm depolarizes before whisking. Therefore, I strongly suggest that it will be included in the main figure rather than as part of the Supplementary figures. I think that it is a 'cleaner' analysis than alignment of the data based on threshold for the detection of whisking onset.

5) Perhaps the authors can do a sanity check for themselves and plot the "Cell average" traces they show in Figure 3 together with 'Whisking average" from the same epochs and see if there is no tendency to whisking signal to appear at the same time as Vm or even before.

6) Moreover, data in some data in Supp Fig. 1 are redundant if I understood these plots correctly: % time whisking in Q and W sum together to 100% so it is enough to show only 1 of them.

7) What is the explanation for the disappearance of depolarization for the dMSN cells of DA-depleted mice? Perhaps more about possible mechanisms should be discussed and also more about the consequence of such change for the behavior.

8) In Supp Fig. 2 a and b it seems that the authors omitted time calibration bar and rely on the x scale of the correlation plot. Although it the same time scale it seems weird to do it.

Reviewer #2 (Remarks to the Author):

The authors have responded constructively to my concerns (and those of the other authors). However, there remain a couple of points that should be clarified.

- The timing of the experiments relative to the 6-OHDA lesion. The authors seem to think that the only relevant variable in the striatal response to 6-OHDA lesioning is dopamine depletion. There are decades of studies showing that this is not true. There are a wide range of cellular and network adaptations that continue to occur well after this phase of the response to the insult has been completed. The authors need to acknowledge this fact. Second, it would be worth taking at least one key independent variable to determine whether it changes significantly within the range of post-survival times actually studied (3-5 wks). Although this latter step would be interesting, it's optional from my standpoint.
- The effects of 6-OHDA lesioning on whisking. I'm sorry but I disagree with the author's conclusions. If the 6-OHDA lesion was complete and includes the striatal regions implicated in whisking, then there are only two mutually exclusive conclusions to be drawn: 1) dopamine and the striatum have no important role in controlling whisking behavior or 2) the basal ganglia circuit involved in whisking undergoes some sort of homeostatic plasticity to keep it operationally intact, despite the loss of dopamine. The authors should add a sentence or two to help the general reader through this section of the Discussion.

Reviewer #3 (Remarks to the Author):

The authors have responded to my comments in a generally satisfactory manner.

I also recognize the great technical difficulty for the in vivo patch-clamp recordings in awake mice and cannot reasonably ask for more experiments.

REVIEWER COMMENTS

Reviewer #1 (Remarks to the Author):

The authors nicely addressed point by point most of my concerns. I now strongly support the paper for publication. However, I would like to see the response of the authors to my minor comments as shown below.

1) I was 'disappointed' to see in Supplementary Figure 1 that whisking parameters were barely different for 6-OHDA mice compared to control mice as I expected to see some effect on behavior - but these are the results. In particular, the lack of change in initiation of whisking when comparing the 6-OHDA mice to control mice.

Indeed, as mentioned by the comment, these are the results we obtained and the impact of the 6-OHDA lesion is not so strong on sensory triggered whisking behavior. There could be multiple reasons for this, but one important aspect of the lesion is that it is unilateral, and therefore the triggering could be mediated by the unlesioned side. Another possibility, also raised by reviewer #2 is that the striatum and striatal dopamine are indeed not instrumental in the onset and offset of whisking behavior and may have a role only in shaping some of the whisking kinetics. There are very few studies using PD models in whisking behavior, and a recent one using a different PD model also showed no measurable effect on whisking behavior (Simanaviciute et al, 2020). We address this point in the discussion [lines 238-242]

2) Yet, did the author find any difference in the duration of whisking following tactile stimulation across these two groups?

No such differences were found.

3) Therefore, I suggest to slightly change the last sentence of the abstract ("Our results show that motor activity affects sensory responses in basal ganglia circuits and that both processes are dopamine and cell type-dependent.") - first change 'motor' to 'whisking' or 'whisking related motor' and also since the whisking parameters were not dopamine dependent it is hard to understand the "both processes are dopamine ..." (the cell type is fine). The authors show that the membrane potential became more depolarized just before whisking. However, it is hard to know from this study if the MSN cells are involved in generation of whisking or just represent motor commands and proprioceptive inputs. Perhaps this should be more emphasized in the discussion.

We now changed the abstract accordingly [lines 22-24] and added the second point in the discussion [lines 239-244]

4) The cross-correlation analysis strongly supports the speculation that Vm depolarizes before whisking. Therefore, I strongly suggest that it will be included in the main figure

rather than as part of the Supplementary figures. I think that it is a 'cleaner' analysis than alignment of the data based on threshold for the detection of whisking onset.

5) Perhaps the authors can do a sanity check for themselves and plot the "Cell average" traces they show in Figure 3 together with "Whisking average" from the same epochs and see if there is no tendency to whisking signal to appear at the same time as Vm or even before.

To address points 4 and 5: We have performed this "sanity" analysis required by the reviewer and added it to Supplementary Fig. 2a. Indeed, it shows that membrane depolarization precedes the whisker movement in both dMSNs and iMSNs. Due to space considerations, we decided to present this new analysis as well as the cross-correlations in Supplementary Fig. 2, which is referred to in the results. Both results are supportive to the main finding, as presented in Fig. 3.

6) Moreover, data in some data in Supp Fig. 1 are redundant if I understood these plots correctly: % time whisking in Q and W sum together to 100% so it is enough to show only 1 of them.

Now changed in the revised Supplementary Fig. 1.

7) What is the explanation for the disappearance of depolarization for the dMSN cells of DA-depleted mice? Perhaps more about possible mechanisms should be discussed and also more about the consequence of such change for the behavior.

The mechanism for this attenuation is not fully clear, however, the experiments and analysis presented in Supplementary Figure 6 suggests that for dMSNs there is an attenuation of motor related signals. This could be due to the lack of dopamine boosting of motor input but may also be a result of network changes in the corticostriatal targets following the lesions.

We can only speculate at this point, which is now added to the discussion [lines 233-234].

8) In Supp Fig. 2 a and b it seems that the authors omitted time calibration bar and rely on the x scale of the correlation plot. Although it the same time scale it seems weird to do it. Changed in revised supp figure.

Reviewer #2 (Remarks to the Author):

The authors have responded constructively to my concerns (and those of the other authors). However, there remain a couple of points that should be clarified.

- The timing of the experiments relative to the 6-OHDA lesion. The authors seem to think that the only relevant variable in the striatal response to 6-OHDA lesioning is dopamine depletion. There are decades of studies showing that this is not true. There are a wide range of cellular and network adaptations that continue to occur well after this phase of the response to the insult has been completed. The authors need to acknowledge this fact. Second, it would be worth taking at least one key independent variable to determine

whether it changes significantly within the range of post-survival times actually studied (3-5 wks). Although this latter step would be interesting, it's optional from my standpoint.

We fully agree with the reviewer that the unilateral 6-OHDA lesion has been studied for decades and there is a cascade of changes, starting from the first day and onward, for months after the lesion. We now clarify this in the revised MS. We definitely do not challenge this fact, however, in this study we chose a specific time point for comparison with unlesioned mice. The experiments presented are challenging and time consuming and we believe it is beyond the scope of this paper to repeat the same processes at different time points after the lesion, in order to study the development of the post-lesion network.

We have now modified the text in order to clarify this point further [lines 239-244].

- The effects of 6-OHDA lesioning on whisking. I'm sorry but I disagree with the author's conclusions. If the 6-OHDA lesion was complete and includes the striatal regions implicated in whisking, then there are only two mutually exclusive conclusions to be drawn: 1) dopamine and the striatum have no important role in controlling whisking behavior or 2) the basal ganglia circuit involved in whisking undergoes some sort of homeostatic plasticity to keep it operationally intact, despite the loss of dopamine. The authors should add a sentence or two to help the general reader through this section of the Discussion.

The relatively small impact of the lesion on whisking parameters indeed indicates a relatively minor role of striatal dopamine on the whisking parameters. It certainly did not abolish whisking or change the kinetic significantly. We have no doubt regarding the severity of the lesion, but it must be emphasized that it is a unilateral lesion and whisking is to a large extent a central behavior, that could be triggered by air puffs to both sides. The impact of PD models on whisking has hardly been studied, however, also in a developmental PD model, very little impact on whisking was observed (Simanaviciute et al, 2020). It could be interesting to study the impact of bilateral lesions and other models for PD.

We have now added this clarification in the revised discussion, as requested by the reviewer [lines 239-244].

Reviewer #3 (Remarks to the Author):

The authors have responded to my comments in a generally satisfactory manner. I also recognize the great technical difficulty for the in vivo patch-clamp recordings in awake mice and cannot reasonably ask for more experiments.

We thank the reviewer for their comments.

REVIEWERS' COMMENTS

Reviewer #1 (Remarks to the Author):

The authors nicely addressed my concerns and therefor I support it for publication as is.